TECHNIQUES AND RESOURCES

# *In vivo* transition in chromatin accessibility during differentiation of deep-layer excitatory neurons in the neocortex

Seishin Sakai[1], Yurie Maeda[1], Mai Saeki[2], Daijiro Konno[3], Keita Kawaji[1], Fumio Matsuzaki[4], Yutaka Suzuki[5], Yukiko Gotoh[1,6,*] and Yusuke Kishi[1,2,*]

## ABSTRACT

During neuronal differentiation, gene transcription patterns change in response to both intrinsic and extrinsic cues. Chromatin regulation at regulatory elements plays a key role in this process. However, how chromatin accessibility evolves *in vivo* in cortical neurons remains unclear. Here, we established a method for labeling differentiating neurons with specific birthdates. Using this method, we traced the 4-day differentiation process of *in vivo* deep-layer excitatory neurons in the mouse embryonic cortex and examined changes in the genome-wide transcription pattern and chromatin accessibility using RNA sequencing and DNase sequencing, respectively. We found that genomic regions of genes linked to mature neuronal functions, including deep layer-specific and stimulus-responsive genes, became accessible even at the embryonic stage. Additionally, our results indicated the involvement of bivalent marks in neural precursor/stem cells and Dmrt3 and Dmrta2 in the regulation of chromatin accessibility during neuronal differentiation. These findings highlight the importance of chromatin regulation in embryonic neurons, enabling the timely activation of neuronal genes during maturation.

KEY WORDS: Cortical neurons, Differentiation, Transcriptome, Chromatin accessibility, Bivalent genes, Dmrt, Mouse

## INTRODUCTION

During cortical development, distinct subtypes of neurons are sequentially generated from common neural precursor/stem cells (NPCs) residing in the ventricular zone (VZ), according to their birthdate (Molyneaux et al., 2007). After neuronal fate commitment, they migrate to the pial surface and constitute a six-layered structure composed of different subtypes of neurons in an inside-out pattern. After completing migration, they mature and acquire neuronal functions, such as neuronal subtype- and layer-specific features and

transcription profiles. For example, layer 6 neurons project their axons to the thalamus and have been implicated as pioneer neurons in callosal projections (De Carlos and O'Leary, 1992; O'Leary and Koester, 1993). Considering that the neuronal subtypes produced by NPCs change daily in the mouse neocortex, it is necessary to label, isolate and analyze specific lineages of neurons in vivo to understand the neuronal differentiation process in a precise manner. Transcriptomic changes during neuronal differentiation of each subtype were revealed using a combination of an original labeling system, FlashTag, and single-cell RNA sequencing (scRNA-seq) (Telley et al., 2016, 2019).

Among the several layers of transcriptional regulation, chromatin accessibility plays a key role in establishing a preparatory state for transcription factors. Transcription factors (TFs) can easily regulate genes in accessible and open chromatin but not in inaccessible and closed chromatin (Boyle et al., 2008b; Crawford et al., 2006). Several studies have shown that chromatin opening during development acts as the basis for the subsequent activation of mature cell types. For instance, regulatory elements determined by the dimethylation of histone H3 at lysine 4 (H3K4me2) are present in intestinal progenitors (Kim et al., 2014). In the hematopoietic system, binding sites of master TFs, such as Foxp3 in regulatory T cells or PU.1 (Spi1) in granulocytes, also became accessible before they are expressed (Luyten et al., 2014; Samstein et al., 2012). Recent single-cell multiome analysis of both gene expression and chromatin accessibility data from the same single cell has also revealed prior opening of chromatin related to lineage specifications during the differentiation process in hair follicles (Ma et al., 2020).

Several studies have demonstrated the role of chromatin accessibility in the developing brain. One pioneering study using DNase sequencing (DNase-seq), a method for profiling chromatin-accessible genomic regions using DNase I, revealed that chromatin opening mediated by Zic transcription factors contributes to transcriptional activation during *in vivo* differentiation of cerebellar granule neurons (Frank et al., 2015). A recent single-cell assay for transposase-accessible chromatin with high-throughput sequencing (ATAC-seq) using developing mouse and human brains also indicated the role of chromatin openness at the regulatory elements of neuronal genes in transcription during neuronal differentiation (Di Bella et al., 2021; Noack et al., 2022; Trevino et al., 2021; Ziffra et al., 2021). However, these studies analyzed all cell types of the developing brain at only one or a small number of stages and assumed a lineage relationship; thus, the changes in chromatin accessibility during the differentiation of a single lineage of neurons remain unknown.

Here, we developed a novel method for genetically labeling a specific lineage of cortical neurons using *in utero* electroporation of wild-type mice and traced the embryonic differentiation of labeled *in vivo* deep-layer neurons. Changes in transcriptome and chromatin accessibility were revealed by simultaneously using RNA sequencing (RNA-seq) and DNase-seq during the differentiation process. We

[1]Laboratory of Molecular Biology, Graduate School of Pharmaceutical Sciences, The University of Tokyo, Tokyo 113-0033, Japan. [2]Laboratory of Molecular Neurobiology, Institute for Quantitative Biosciences, The University of Tokyo, Tokyo 113-0032, Japan. [3]Cellular and Molecular Bioengineering Laboratory, Graduate School of Science and Engineering, Kindai University, Osaka 577-8502, Japan. [4]Department of Aging Science and Medicine, Graduate School of Medicine, Kyoto University, Kyoto 606-8507, Japan. [5]Department of Computational Biology, Graduate School of Frontier Sciences, The University of Tokyo, Chiba 277-8561, Japan. [6]International Research Center for Neurointelligence (WPI-IRCN), The University of Tokyo, Tokyo 113-0033, Japan.

*Authors for correspondence (ygotoh@mol.f.u-tokyo.ac.jp; ykisi@iqb.u-tokyo.ac.jp)

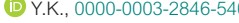 Y.K., 0000-0003-2846-5400

Handling editor: James Briscoe

found that genes with promoter regions that opened during neuronal differentiation in the embryonic stage were enriched in deep layer-specific genes and genes responsive to brain-derived neurotrophic factor (BDNF) stimulation, indicating that the regulation of chromatin accessibility might contribute to acquisition of a preparatory state for the induction of genes related to mature neuronal functions. Furthermore, we also found that chromatin opening during embryonic neuronal differentiation occurred mainly in bivalent genes in NPCs that harbored both H3K4me3 and H3K27me3 histone modifications and that the transcriptional repressors Dmrt3 and Dmrta2 were involved in the activation of opening genes.

## RESULTS

### *In vivo* tracing of differentiation in neurons with a specific birthdate in the neocortex using genetic labeling with *in utero* electroporation

To understand neuronal differentiation in the neocortex, we first attempted to genetically label a specific lineage of neurons determined by their birthdate, as neocortical NPCs sequentially produce different types of neurons at the embryonic stage. We introduced two plasmids, pAAV-DIO-GFP and pNeuroD1-ERT2CreERT2, into the NPCs via *in utero* electroporation. pAAV-DIO-GFP expresses GFP after the activation of the Cre recombinase because the GFP cassette is inversely inserted between the two types of loxP sequences (Schnütgen et al., 2003). As neurons transiently express *NeuroD1* in their immature state after cell division (Miyoshi and Fishell, 2012), pNeuroD1-ERT2CreERT2 expresses tamoxifen-activating Cre recombinase only in NeuroD1-positive immature neurons (Guerrier et al., 2009). Therefore, we permanently labeled a specific lineage of immature neurons with the same birthdate when tamoxifen was injected (Fig. 1A). To validate this method, we introduced these two plasmids together with an mCherry-expressing plasmid under the control of the ubiquitous promoter CAG as a transfection control into embryonic day (E) 12.0 neocortex and injected tamoxifen at E13.0 (Fig. 1B). Immunostaining of embryos at E16.0 showed that GFP-labeled cells labeled with Ctip2 (Bcl11b), a marker for deep-layer neurons (Arlotta et al., 2005), were mainly located in the deep layer. In contrast, mCherry-positive cells were distributed not only in the cortical plate (CP), but also in the intermediate zone (IMZ), and even in the ventricular and subventricular zones (VZ/SVZ). This suggested that neurons produced at a later stage, at least 4 days after transfection, were also labeled with a plasmid containing a ubiquitous promoter. In contrast, our tracing method specifically labeled Ctip2-positive deep-layer neurons produced before tamoxifen injection.

Next, we confirmed the differentiation status of the labeled neurons at E13.0 and E14.0. Since tamoxifen injection had not yet been performed at E13.0, we visualized neurons labeled by transfecting pNeuroD1-IRES-GFP (Fig. 1C) and found that GFP-positive cells were mainly located in the IMZ. At E14.0, 1 day after tamoxifen injection, the GFP-positive neurons were primarily retained in the IMZ, although some moved to the CP (Fig. 1D). Therefore, this labeling method allowed us to trace the differentiation process of deep-layer neurons at a temporal resolution of 1 day.

To examine their gene expression patterns, we next isolated these cells using fluorescence-activated cell sorting (FACS) at E14.0 and E16.0, together with CD133 (Prom1)-high NPCs at E12.0, as progenitor cells for labeled cells. pNeuroD1-IRES-GFP-positive cells were sorted at E13.0 as neurons that were labeled by tamoxifen injection in our system (Fig. 2A). Reverse transcription and quantitative polymerase chain reaction (RT-qPCR) analysis confirmed that the expression level of *Nes* (encoding nestin), an NPC marker, was reduced, whereas those of *Tubb3*, *Gabrb2* and

neuronal markers were increased in pNeuroD1-ERT2CreERT2-labeled cells at E16.0 compared to CD133-high cells at E12.0 or pNeuroD1-IRES-GFP-positive cells at E13.0 (Fig. 2B). Consistent with pNeuroD1 activity, endogenous *NeuroD1* was highly expressed in pNeuroD1-IRES-GFP-positive cells at E13.0. These data suggest that embryonic differentiation of deep-layer excitatory neurons can be successfully traced using our labeling system. CD133-high cells at E12.0, pNeuroD1-IRES-GFP-positive cells at E13.0, and pNeuroD1-ERT2CreERT2-labeled cells at E14.0 and E16.0 are referred to as E12.0 NPCs, E13.0 neurons, E14.0 neurons and E16.0 neurons, respectively.

Next, we performed RNA-seq analysis using E12.0 NPCs and E13.0, E14.0 and E16.0 neurons. Developmentally regulated genes during differentiation from E12.0 NPCs to E16.0 neurons were determined and classified into nine clusters using maSigPro software (FDR<0.05) (Conesa et al., 2006). The transcription levels of more than half of the expressed protein-coding genes (8403 out of 14,286 expressed genes) changed significantly (Fig. 2C, Table S1), indicating global changes in transcription patterns after 4 days of differentiation from NPCs to embryonic neurons. Genes related to neuronal functions, such as synaptic transmission, were specifically enriched (cluster 2) (Fig. 2C). In contrast, genes related to the cell cycle were enriched in the downregulated genes (cluster 6), indicating that cell cycle exit occurs at the onset of neuronal differentiation. Notably, genes related to lipid biosynthesis were specifically enriched in clusters of temporally suppressed genes (cluster 9) (Fig. 2C). This suggests that the synthesis of lipids, essential components of the membrane, is important for the future maturation of neurons, such as elaboration of neurites, and for supplying lipids for cell cycle-associated increases in the plasma membrane. Overall, these results suggest a dynamic change in transcription states during only 4 days of differentiation from NPCs in the embryonic stage *in vivo*.

### Changes in DNase-hypersensitive sites during *in vivo* differentiation of deep-layer neurons

Since regulation of chromatin accessibility at gene regulatory sites is important for regulation of the transcription states of target genes, we next performed DNase-seq using E12.0 NPCs and E13.0, E14.0 and E16.0 neurons (Fig. 3A). Isolated nuclei were digested using DNase I, and DNase I-sensitive fragments shorter than 1 kbp were collected by gel extraction after agarose gel electrophoresis at each stage. The sequence of each collected DNA fragment was determined using high-throughput sequencing and DNase-hypersensitive sites (DHSs) were determined using F-Seq software (Boyle et al., 2008a) (Fig. 3B). In our analysis, approximately 20,000-30,000 DHSs were determined at each stage, and E16.0 neurons had the most DHSs compared to those in other stages. The total width of the DHSs was higher in E12.0 NPCs and E16.0 neurons than in E13.0 and E14.0 neurons (Fig. 3B), suggesting wider DHSs in E12.0 NPCs. To confirm whether DNase-seq appropriately identified open chromatin regions, we examined the distribution of DHSs in the genome and the relationship between the openness of gene loci and their expression levels, as determined using RNA-seq. Consistent with previous reports (Boyle et al., 2008b), DHSs determined using our DNase-seq were enriched in promoter regions, defined by −1 kb to +0.5 kb of transcription start sites, and exon regions compared to their proportion in the whole genome (Fig. 3C). We then identified genes with at least one DHS in their promoters as DHS positive and those without any DHSs in their promoters as DHS negative. As expected, the transcription levels of DHS-positive genes were higher than those of DHS-negative genes at all differentiation stages (Fig. 3D).

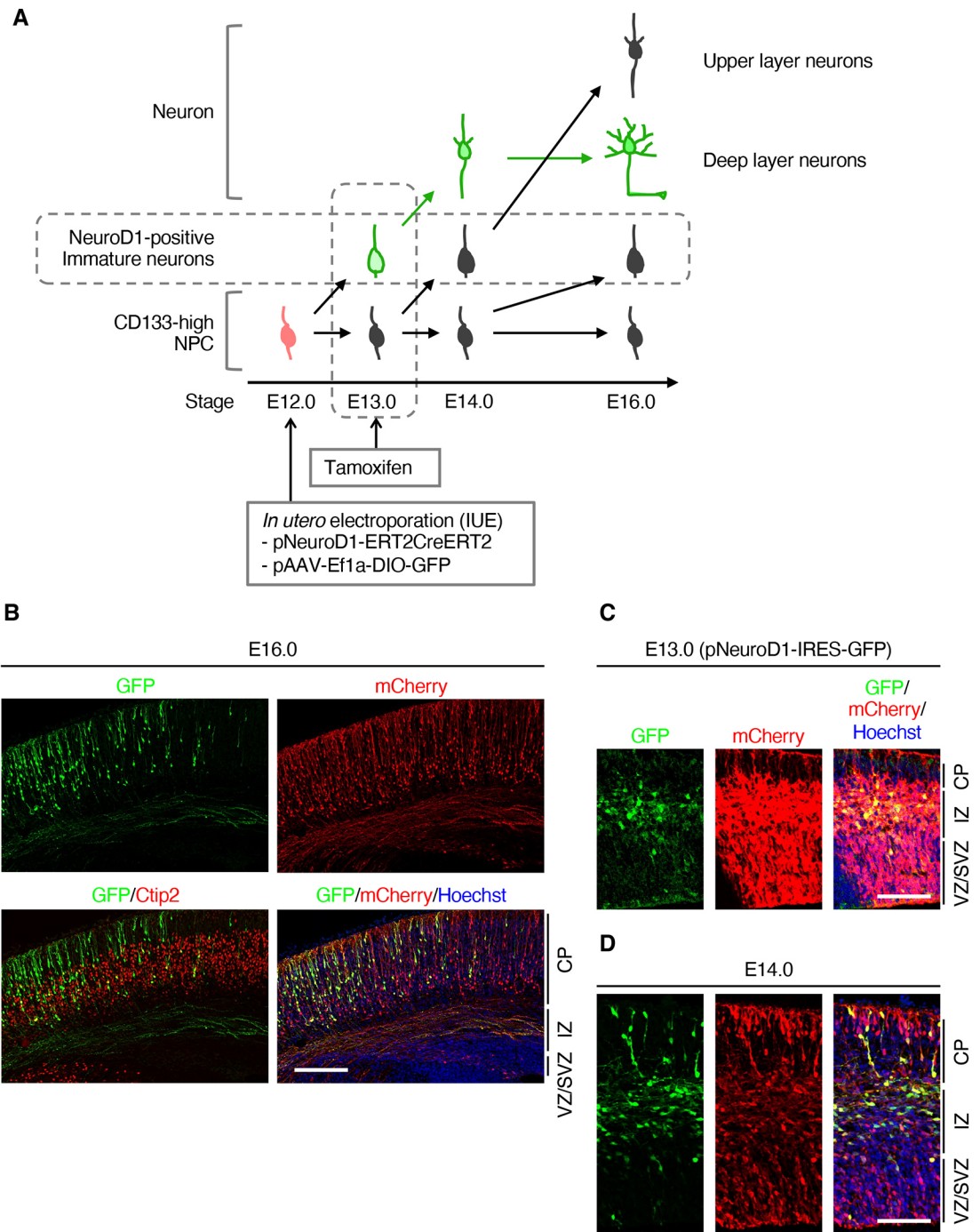

**Fig. 1. Genetic labeling of deep-layer neurons using *in utero* electroporation.** (A) Experimental scheme used to label deep-layer neurons and trace their differentiation process. (B-D) pNeuroD1-ERT2CreERT2, pAAV-Ef1a-DIO-GFP and pCAG-mCherry plasmids (B,D) or pNeuroD1-IRES-GFP and pCAG-mCherry plasmids (C) were injected into the lateral ventricle of E12.0 embryos and electroporated. Pregnant mice were injected with tamoxifen at E13.0 (B,D). The brains were dissected out from the uterus at indicated stages and subjected to immunohistochemistry with anti-GFP, anti-RFP (mCherry) and anti-Ctip2 (B) antibodies. Nuclei were counterstained with Hoechst 33342. Images are representative of three samples. Scale bars: 400 µm (B); 200 µm (C, D). CP, cortical plate; IZ, intermediate zone; VZ/SVZ, ventricular/subventricular zone.

These results suggested that DNase-seq determined accessible chromatin regions that are enriched in gene-coding regions, especially in promoter regions and active gene loci, as previously reported (Boyle et al., 2008b).

Using DNase-seq data, we aimed to elucidate the significance of chromatin accessibility transitions during *in vivo* differentiation of deep-layer neurons. First, we analyzed the relationship between gene expression changes and DHS dynamics. Genes were annotated based on their promoter or enhancer regions, as defined by the EnhancerAtlas 2.0 (Gao and Qian, 2019a), and categorized into 15 gene sets based on the presence or absence of DHSs at each stage (Table S1). We then examined the overlap between the nine differentially expressed gene (DEG) clusters shown in Fig. 2C and the DHS gene sets, assessing statistical significance using Fisher's exact test (Fig. 4A).

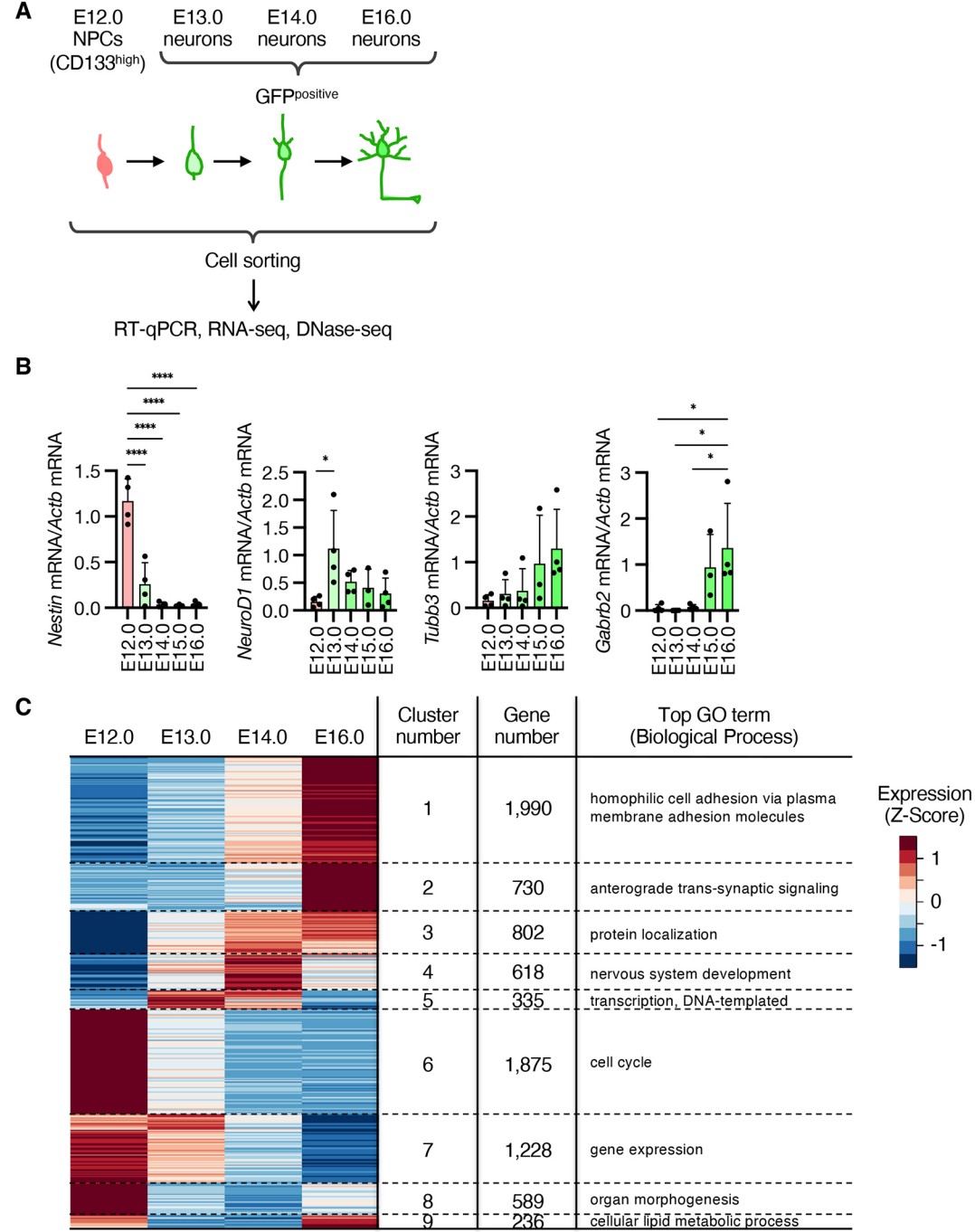

**Fig. 2. Transcriptomic changes during 4-day differentiation of deep-layer neurons.** (A) Experimental scheme to isolate E12.0 NPCs and GFP-labeled neurons at E13.0, E14.0 and E16. (B) RT-qPCR analysis of isolated NPCs and neurons. Relative abundance of the indicated mRNA (normalized to the amount of *Actb* mRNA) is shown. Data are shown as mean+s.d. from three (E15.0) or four (E12.0, E13.0, E14.0 and E16.0) independent experiments. *P*-values between five groups were determined by one-way ANOVA followed by Tukey's multiple comparison test. *$P<0.05$, ****$P<0.0001$. (C) RNA-seq analysis of isolated E12.0 NPCs and E13.0, E14.0 and E16.0 neurons. Expression levels of DEGs between four cell types are shown in the heatmap as Z-score values. The DEGs were categorized into nine clusters, and the gene numbers and the top GO term in the Biological Process for each gene cluster are shown. Data are from two independent experiments.

Focusing on enhancer regions, we found that genes with enhancers that remained open at all stages were specifically enriched in the DEG clusters (Fig. 4A, right), suggesting that enhancers contribute to changes in gene expression via mechanisms other than alterations in chromatin accessibility. In contrast, genes that acquired DHSs at their promoters during *in vivo* differentiation, particularly those that opened exclusively in E16 neurons, were significantly enriched in the

upregulated DEGs (Fig. 4A, left; Fig. 4B). For example, *Rbfox1*, a gene expressed in differentiated neurons (Gehman et al., 2011), exhibited higher DNase-seq signals around its promoter regions and increased expression levels (Fig. 4C). Compared to closed regions, open chromatin regions in E12.0 NPCs to E16.0 neurons were specifically enriched in promoter regions (Fig. 3E). This is consistent with previous reports showing that neuron-specific accessible regions

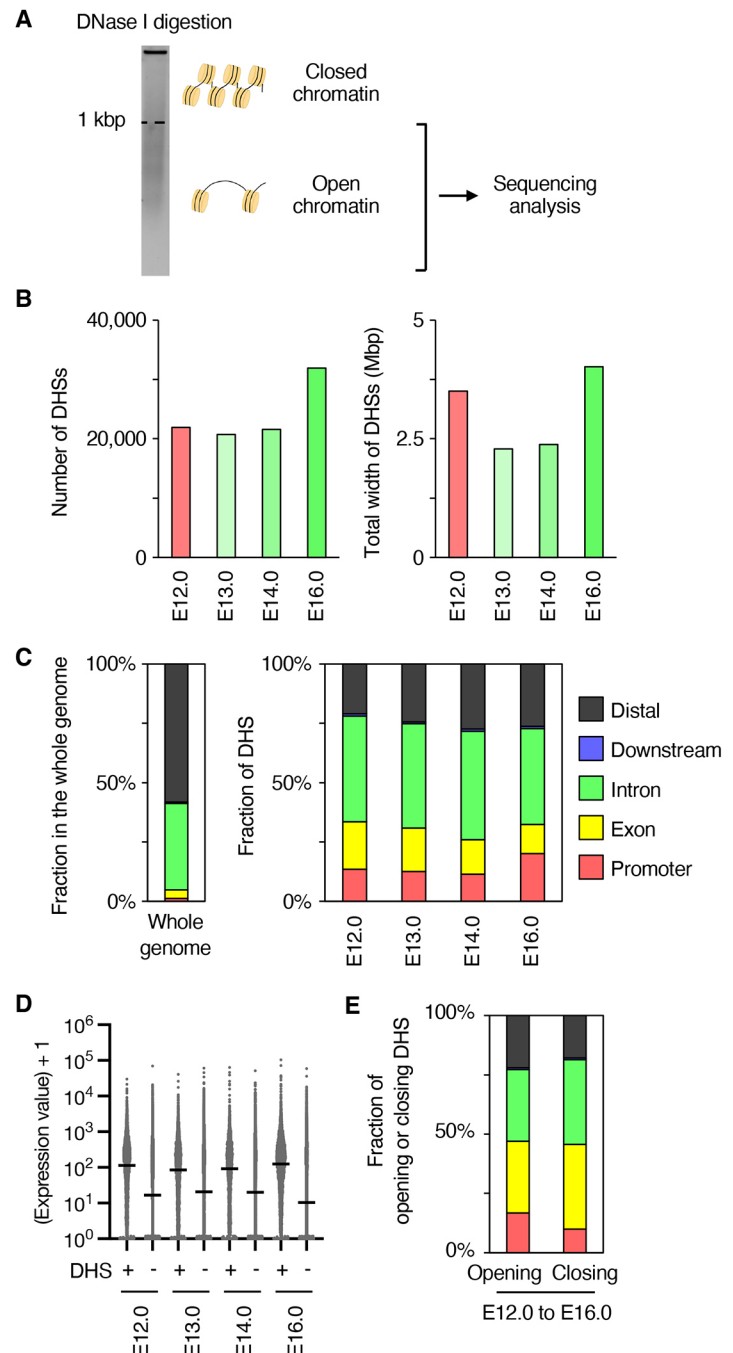

**Fig. 3. Changes in chromatin accessibility during 4-day differentiation of deep-layer neurons.** (A) Experimental scheme of DNase-seq analysis. Data are from two independent experiments. (B) The numbers (left) and total width (right) of DHSs were identified at each time point. (C) Fractions of genomic features in the whole mouse genome (left) and DHSs at each time point (right). (D) Expression levels of genes with or without DHSs at each time point. Horizontal black lines mark the median. (E) Fractions of genomic features in DHSs of opening or closing genes from E12.0 NPCs to E16.0 neurons.

are enriched in promoters (de la Torre-Ubieta et al., 2018). We also found that the expression levels of genes with DHSs in their promoter regions only at E16.0 significantly increased at E16.0, compared to other stages (Fig. 4D). These results suggest that chromatin accessibility plays a crucial role in regulating gene expression changes, particularly gene upregulation, during *in vivo* differentiation, primarily in promoter regions.

Gene Ontology (GO) analysis of stage-specific open genes revealed the enrichment of genes related to cellular stress and RNA processing at E12.0 (Table S2). In contrast, the genes that became accessible only at E16.0 were associated with neural development, synapse formation and dendritic function (Fig. 4E, Table S2), indicating that chromatin opening contributes to the acquisition of neuronal functions. Additionally, Kyoto Encyclopedia of Genes

and Genomes (KEGG) pathway analysis identified enrichment not only in neural developmental pathways, such as axon guidance and neurotrophin signaling, but also in pathways related to neurological disorders (Fig. 4E).

To investigate the regulatory mechanisms further, we performed motif analysis using HOMER and TF binding analysis using the ChIP-Atlas, a database of transcription factors and chromatin regulator-binding sites derived from public chromatin immunoprecipitation (ChIP)-seq data (Oki et al., 2018) (Tables S3-S6). In both analyses, the specific DHSs of E16 neurons exhibited the highest enrichment of motifs and TFs with lower *P*-values, whereas the specific DHSs of E13 and E14 neurons showed few or no significantly enriched motifs and TFs. As expected, E12.0-specific DHSs were enriched for Ascl1 and

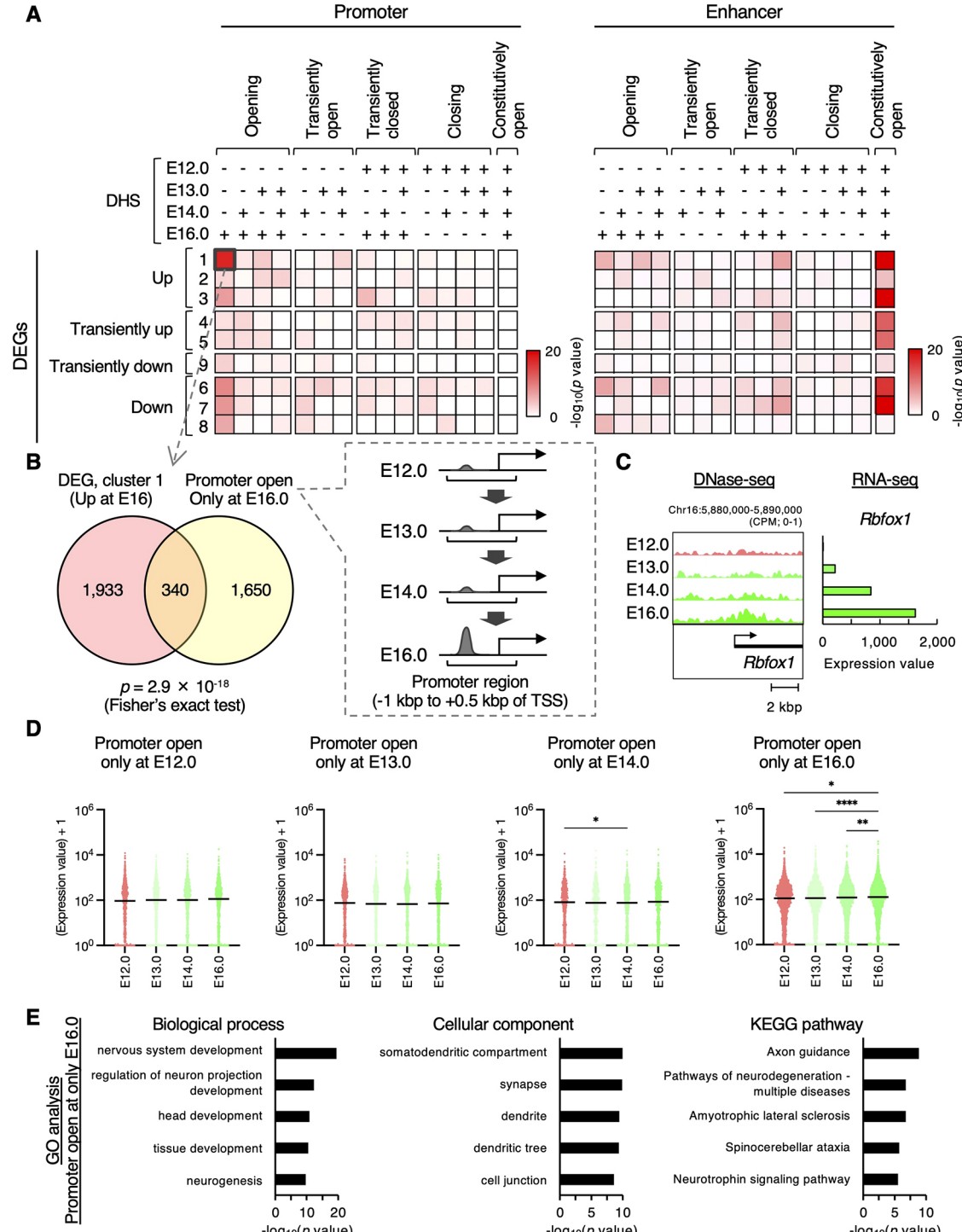

**Fig. 4. Chromatin opening at promoter regions of neuronal genes during 4-day differentiation of deep-layer neurons.** (A) Significance of the overlap between DEG clusters (as shown in Fig. 2) and genes with (+) or without (−) DHSs at their promoter (left) or enhancer (right) regions was determined using Fisher's exact test. (B) Venn diagram showing an example of the overlap between DEG cluster 1 and the genes with DHSs at their promoter regions only at E16.0. (C) DNase-seq signals and expression levels of *Rbfox1* genes in DEG cluster 1 with DHSs at its promoter region only at E16.0. (D) Expression levels of genes with DHSs at their promoter regions at each stage are shown. *P*-values were determined using the Friedman test, followed by Dunn's multiple comparison test. *$P<0.05$, **$P<0.01$, ****$P<0.0001$. Horizontal black lines mark the median. Data are from two independent experiments. (E) The top ten GO terms and the *P*-values of genes with DHSs at their promoter regions at only E16.0.

Neurog2 motifs (HOMER, known motif, $q<0.01$), whereas E16.0-specific DHSs were bound by Tbr1, Foxp1, Neurod1 and Neurod2 [HOMER, known motif, $q<0.01$; ChIP-Atlas, Neural type, fold enrichment (FE)>1], which are well-known TFs for the function

of NPCs and neurons, respectively (Tables S3, S6) (Arnold et al., 2008; Hevner et al., 2006; Li et al., 2015; Olson et al., 2001; Sessa et al., 2008). Furthermore, well-known risk factors for neurodevelopmental disorders, Chd8 and Setd1a (ChIP-Atlas,

Neural type, FE>1), were also identified in the E16-specific DHSs (Table S6) (Bernier et al., 2014; Takata et al., 2014). These findings highlight the involvement of chromatin accessibility transitions in neuronal differentiation and development. Taken together, these data on the transition of chromatin accessibility during the initial phase of neuronal differentiation *in vivo* indicate its contribution to proper neural developmental processes.

### Acquiring chromatin accessibility during embryonic differentiation for the preparatory state of neuronal gene loci

Chromatin accessibility contributes not only to transcriptional activity but also to the potential for future induction of genes. Indeed, previous reports have found an increase in chromatin accessibility in gene loci that are activated or bound by TFs during the developmental process of neural and non-neural systems (Frank et al., 2015; Kim et al., 2014; Luyten et al., 2014; Samstein et al., 2012). To examine whether the regulation of chromatin accessibility contributes to the induction of genes in deep-layer excitatory neurons, we examined the chromatin state of neuronal gene sets that were mostly activated in the postnatal and adult stages. In particular, we focused on layer-specific genes and genes responsive to external stimuli.

Cortical neurons have layer-specific functions and specific morphologies and target brain regions. Deep-layer neurons, including layer 5 and 6 neurons, the main layers analyzed in this study, also perform specific functions (Arlotta et al., 2005; De Carlos and O'Leary, 1992; Molyneaux et al., 2007; O'Leary and Koester, 1993). To examine the chromatin accessibility of layer-specific genes, we utilized a previously reported RNA-seq analysis (Fig. 5A) (Belgard et al., 2011) and investigated the enrichment of opening and closing genes from E12.0 NPCs to E16.0 neurons using Gene Set Enrichment Analysis (GSEA) (Mootha et al., 2003; Subramanian et al., 2005). The opening genes were significantly enriched in layer 6-specific genes, whereas the closing genes were enriched in layer 2/3-specific genes (Fig. 5B, Table S1). We also observed a significant overlap between layer 6-specific genes and genes with promoter opening at E16.0, as well as between layer 2/3-specific genes and genes with promoter closure from E12.0 to E16.0 (Fig. S1). Furthermore, motif and ChIP-Atlas analyses revealed enrichment of the layer 6-specific TFs Tbr1 and Foxp1 in specific DHSs of E16 neurons, whereas the layer 2/3-specific TF Cux2 was enriched in DHSs of E12 NPCs (HOMER, known motif, q<0.01; ChIP-Atlas, Neural type, FE>1) (Tables S3, S6) (Cubelos et al., 2010; Hevner et al., 2001; Pearson et al., 2020). Considering that isolated neurons mostly differentiate into deep-layer excitatory neurons (Fig. 1B), these data suggest that the preparatory state for deep-layer-specific gene expression patterns is already established during embryonic differentiation. In addition, the expression levels of deep layer-specific genes that opened during differentiation from E12.0 NPCs to E16.0 neurons was slightly increased in E16.0 neurons compared with E12.0 NPCs, whereas they were obviously upregulated in layer 6 neurons at postnatal day (P) 56 compared with E16.0 neurons (Fig. 5C) (Belgard et al., 2011). This further supports the idea that a preparatory state for deep layer-specific gene expression patterns is already established at the embryonic stage.

Neuronal activity induced by external stimuli, especially during the postnatal stage, is important for the functional maturation of neurons as they can regulate neurite extension and synapse formation (West and Greenberg, 2011). The expression of genes that respond to external stimuli, such as neurotrophic factors, plays a crucial role in this process. Therefore, we focused on genes that respond to external stimuli. To define these gene sets, we performed a microarray analysis of BDNF-stimulated primary neurons *in vitro*. *In vitro* neurons cultured for 6 days from the E15 cortex were stimulated with BDNF for 1 h. Using the results of the microarray, GSEA analysis for genes opening from E12.0 NPCs to E16.0 neurons showed significant enrichment in genes upregulated by BDNF stimulation (Fig. 5E, Table S7). We also observed enrichment of motifs and binding sites of responsive TFs, including Fos, Fra2 (Fosl2) and JunB, in the AP-1 complex, as well as Mef2 and NPAS family proteins, as determined using motif and ChIP-Atlas analyses (HOMER, known motif, q<0.01; ChIP-Atlas, Neural type, FE>1) (Tables S3, S6) (Yap and Greenberg, 2018). Furthermore, genes showing both opening during differentiation and upregulation after BDNF stimulation specifically included those related to neuronal development, as determined by GO analysis (Fig. 5F). Genes closing from E12.0 NPCs to E16.0 neurons were also enriched in genes upregulated by BDNF stimulation (Fig. 5E), including those related to metabolic processes (Fig. 5F). Overall, these results support the notion that chromatin opening during embryonic differentiation permits the induction of genes associated with neuronal functions, such as layer-specific genes, and those responsive to neurotrophic factors.

### Opening of bivalent gene loci in NPCs during neuronal differentiation

What regulates the transition in chromatin accessibility, particularly chromatin opening, during neuronal differentiation? Upon re-examining the ChIP-Atlas results for specific DHSs of E16 neurons, we observed the enrichment of polycomb group (PcG) proteins, including Rnf2 (also known as Ring1B), Suz12 and Ezh2, in pluripotent stem cell types (Schuettengruber et al., 2017) (Fig. 6A, Table S6). Additionally, the trithorax group (TrxG)-related proteins Brd4 and Ash2l were enriched in neural cell types. In embryonic stem cells (ESCs), bivalent genes that harbor both H3K4me3 (modified by TrxG) and H3K27me3 (modified by PcG) marks are known to be activated or inactivated during subsequent differentiation into specific lineages (Azuara et al., 2006; Bernstein et al., 2006; Zhao et al., 2007). Since the properties of E12.0 NPCs and ESCs are similar, we hypothesized that the bivalent state of E12.0 NPCs plays a crucial role in facilitating chromatin opening during their differentiation into E16.0 neurons.

Therefore, to examine the chromatin regulation of bivalent genes in NPCs, we determined the bivalent gene set in E12.0 NPCs using ChIP-seq analysis for H3K4me3 and H3K27me3 (Fig. 6B). Peak calling of H3K4me3 and H3K27me3 revealed 10,711 H3K4me3$^+$, 862 H3K27me3$^+$, and 629 H3K4me3$^+$ and H3K27me3$^+$ bivalent genes in their promoter regions (Fig. 6B, Table S1). Consistent with the results of the ChIP-Atlas analysis (Fig. 6A), a significant portion of bivalent genes in NPCs became open during neuronal differentiation (214 of 629 bivalent genes) (Fig. 6C). Notably, among four gene sets, the expression levels of H3K4me3$^+$ H3K27me3$^+$ bivalent genes specifically increased during neuronal differentiation, whereas those of the other gene sets (H3K4me3$^-$ H3K27me3$^-$, H3K4me3$^+$ H3K27me3$^-$ and H3K4me3$^+$ H3K27me3$^-$) exhibited either no increase or a slight reduction (Fig. 6D). This observation was further supported by the specific enrichment of H3K4me3$^+$ H3K27me3$^+$ bivalent genes among the upregulated DEGs (Fig. S2). Additionally, GO analysis revealed that bivalent genes in NPCs were enriched for terms associated with neural development and mature neuronal functions, such as synapse formation and neuron projection (Fig. 6E). Overall, these results suggested that the opening of bivalent states in E12.0 NPCs contributes to the upregulation of neuronal genes during embryonic neuronal differentiation.

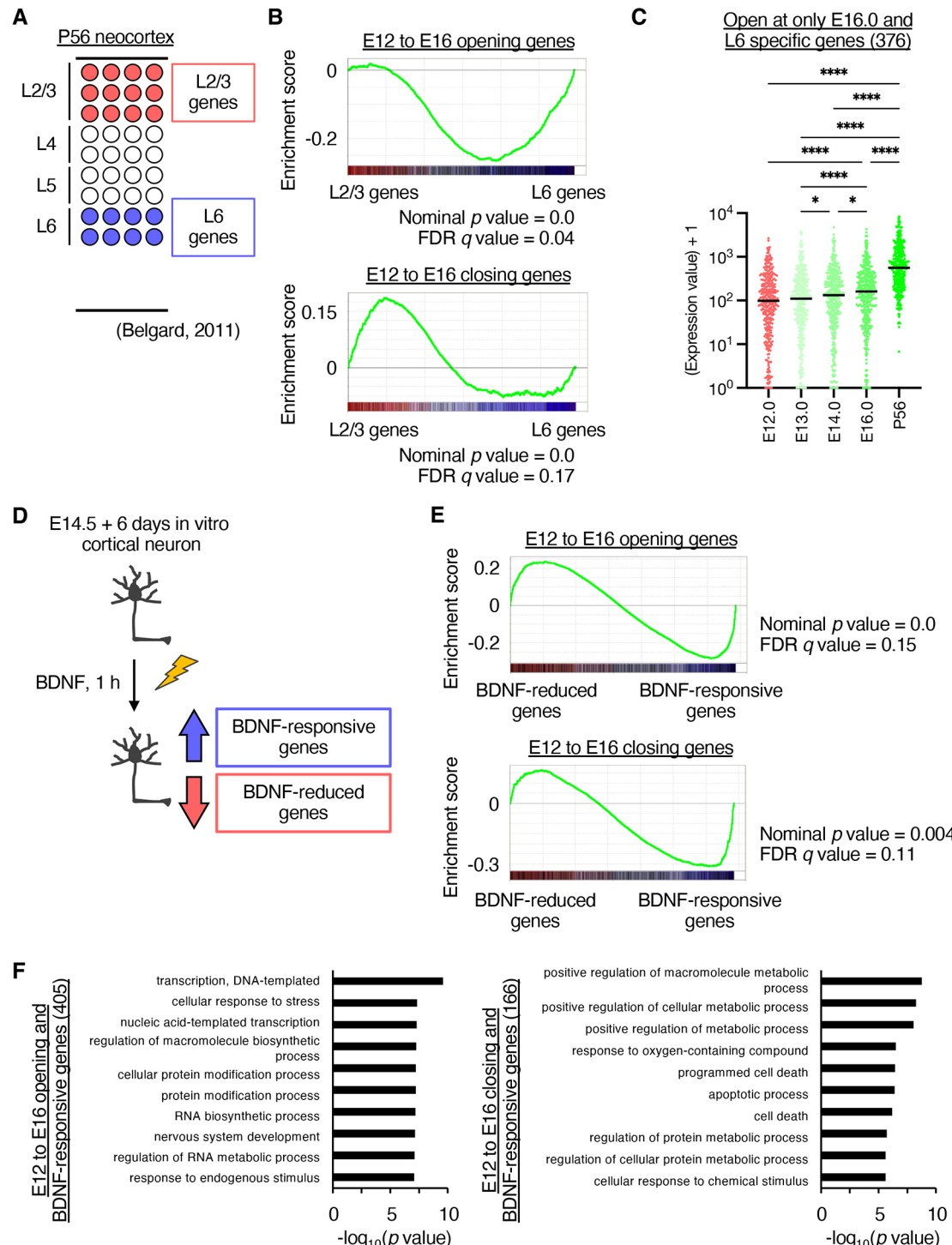

**Fig. 5. Association of genes opening from E12.0 NPCs to E16.0 neurons and deep layer-specific or BDNF-responsive genes.** (A) RNA-seq analysis of layer 2/3 (upper layer) and layer 6 (deep layer) neurons at P56 (Belgard et al., 2011). (B) GSEA of opening (upper panel) or closing (lower panel) genes from E12.0 NPCs to E16.0 neurons in RNA-seq of layers 2/3 and 6. (C) Expression levels of opening and layer 6-specific genes from E12.0 NPCs to E16.0 neurons during differentiation from E12.0 NPCs to P56 neurons. *P*-values were determined using the Friedman test, followed by Dunn's multiple comparison test. *$P<0.05$, ****$P<0.0001$. Horizontal black lines mark the median. $n=1$. (D) Microarray analysis of *in vitro* primary culture neurons with or without BDNF stimulation for 1 h. Data are from three independent experiments. (E) GSEA of opening (upper panel) or closing (lower panel) genes from E12.0 NPCs to E16.0 *in vitro* primary culture neurons with or without BDNF stimulation. (F) GO analysis of opening (left) or closing (right) genes and BDNF-responsive genes from E12.0 NPCs to E16.0 neurons.

## Gene activation due to reduction of Dmrt3 and Dmrta2 during neuronal differentiation

Next, we investigated the molecular mechanisms underlying chromatin opening during neuronal differentiation. Among the TFs for which motifs were enriched in the DHSs of E16 neurons, we focused on Dmrt3 (Fig. 7A, Table S3). Dmrt3 and its family member, Dmrta2 (also known as Dmrt5), share recognition DNA motifs and act as transcriptional repressors (Desmaris et al., 2018; Konno et al.,

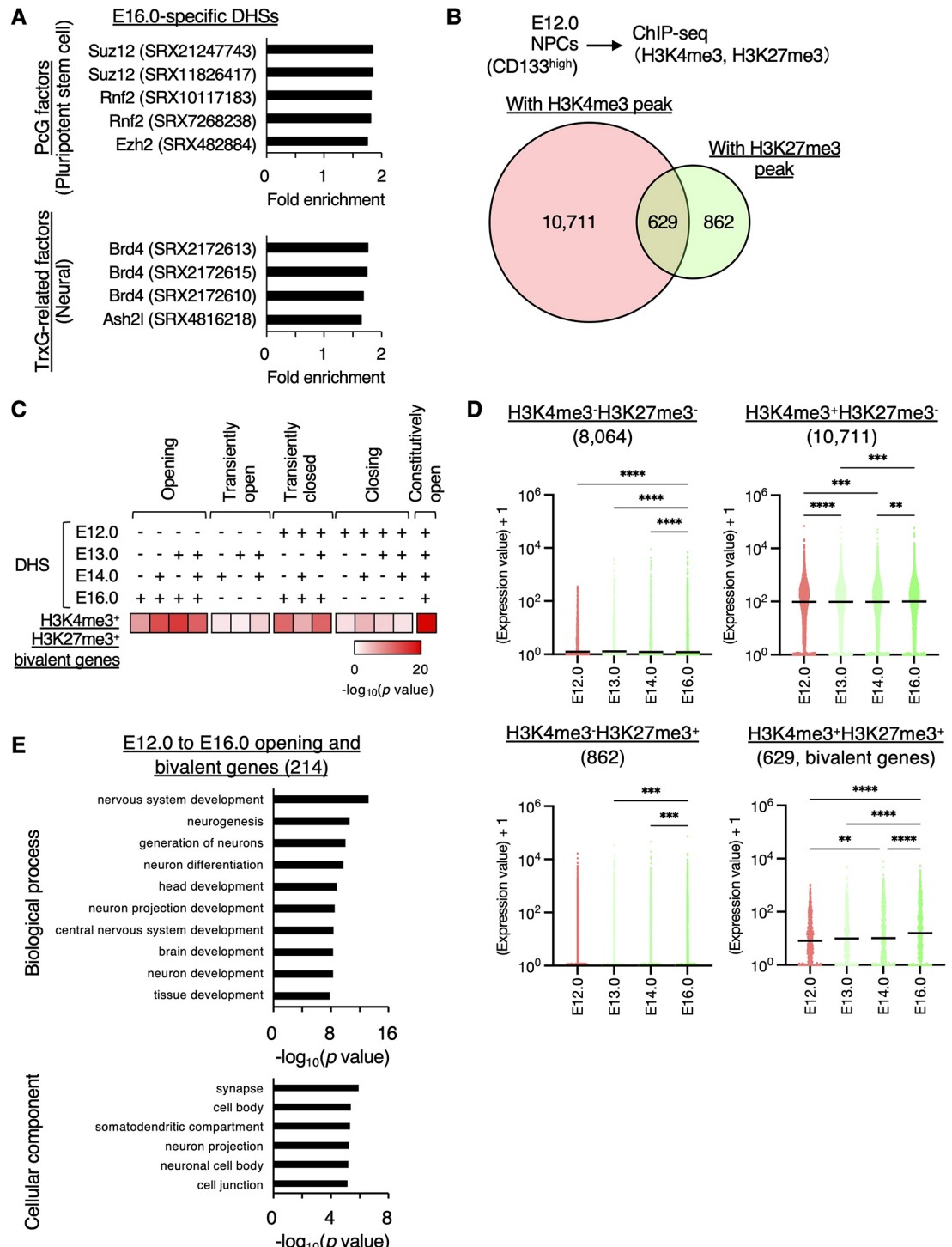

**Fig. 6. Association between chromatin opening during neuronal differentiation and bivalent state in NPCs.** (A) PcG or TrxG-related factors enriched in E16.0-specific DHSs and their fold enrichment were determined using ChIP-Atlas analysis. (B) E12.0 NPCs were subjected to ChIP-seq analysis with H3K4me3 and H3K27me3 antibodies. Genes with both H3K4me3 and H3K27me3 peaks at their promoter regions were identified. Data are from two independent experiments. (C) The significance of the overlap between bivalent genes and genes with (+) or without (−) DHSs at their promoter regions was determined using Fisher's exact test. (D) The expression levels of H3K4me3− H3K27me3−, H3K4me3+ H3K27me3−, H3K4me3− H3K27me3+ and H3K4me3+ H3K27me3+ bivalent genes in E12.0 NPCs during 4-day differentiation are shown. P-values were determined using the Friedman test, followed by Dunn's multiple comparison test. **$P<0.01$, ***$P<0.001$, ****$P<0.0001$. Horizontal black lines mark the median. Data are from two independent experiments. (E) GO analysis of opening and bivalent genes from E12.0 NPCs to E16.0 neurons.

2019; Murphy et al., 2007). Both these proteins are highly expressed in NPCs and play crucial roles in determining dorsal NPC identity (Desmaris et al., 2018; Konno et al., 2019). Knockout of both Dmrt3

and Dmrta2 induces premature neurogenesis (Konno et al., 2019), suggesting their roles in maintaining the undifferentiated state of NPCs. Based on this, we hypothesized that a reduction in Dmrt3 and

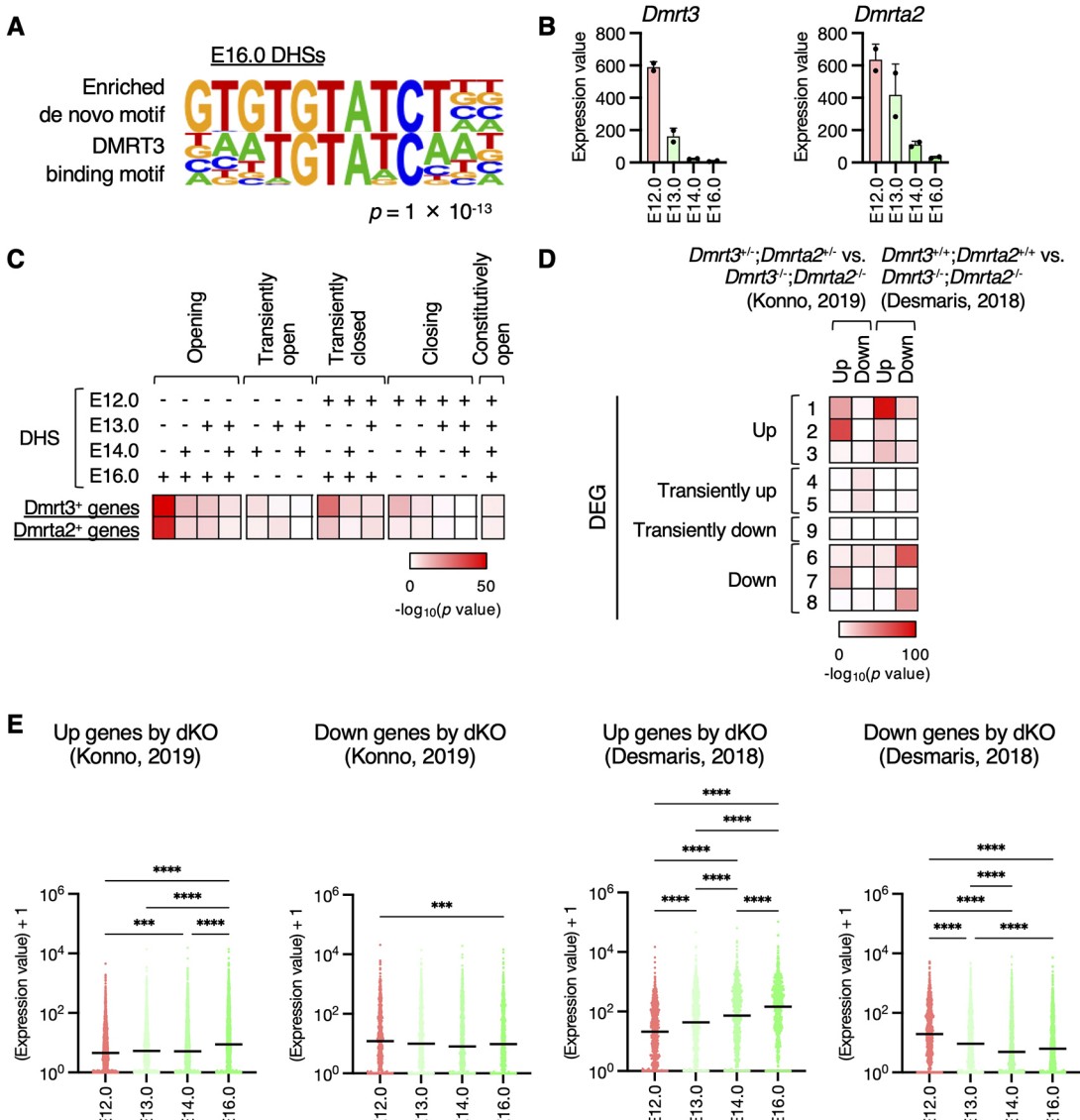

**Fig. 7. Association between Dmrt family proteins and gene activation during neuronal differentiation.** (A) Motif analysis for E16.0 DHSs using HOMER software. The enriched motif (top) and Dmrt3-binding motif (bottom) are shown. (B) Expression levels of *Dmrt3* and *Dmrta2* were determined using RNA-seq as described for Fig. 2. (C) Significance of the overlap between genes with Dmrt3 or Dmrta2 peaks at their promoter regions and genes with (+) or without (−) DHSs at their promoter regions was determined using Fisher's exact test. (D) Significance of the overlap between DEG clusters determined in Fig. 2 and the up- or downregulated genes obtained by knocking out both *Dmrt3* and *Dmrta2* in the E12.5 telencephalon was determined using the Fisher's exact test. (E) Expression levels of up- or downregulated genes in *Dmrt3* and *Dmrta2* double knockout E12.5 telencephalon are shown. *P*-values were determined using the Friedman test, followed by Dunn's multiple comparison test. \*\*\**P*<0.001, \*\*\*\**P*<0.0001. Horizontal black lines mark the median. Data are from two independent experiments.

Dmrta2 levels contributes to chromatin opening and gene activation during neuronal differentiation.

First, we confirmed the higher expression of *Dmrt3* and *Dmrta2* in E12.0 NPCs compared to that in differentiated neurons in the results of our RNA-seq analysis (Fig. 7B). By analyzing ChIP-seq data for Dmrt3 and Dmrta2 in the E12.5 telencephalon (Konno et al., 2019), we identified genes with Dmrt3 or Dmrta2 peaks in their promoter regions. Notably, there was significant overlap between opening genes, particularly those that opened at E16, and Dmrt3- or Dmrta2-positive genes (Fig. 7C, Table S1). Additionally, we found that the 145 Dmrt3-positive genes were bivalent ($P=1.9\times10^{-7}$, Fisher's exact test). Considering the function of Dmrt3 and Dmrta2 as transcriptional repressors, these results suggested that the reduction in Dmrt3 and Dmrta2 levels from E12.0 NPCs to E16.0 neurons induces chromatin opening during neuronal differentiation.

To investigate further whether Dmrt3 and Dmrta2 repress neuronal genes in NPCs, we analyzed RNA-seq data from the E12.5 telencephalon of *Dmrt3* and *Dmrta2* double-knockout mice [$Dmrt3^{-/-};Dmrta2^{-/-}$ compared to $Dmrt3^{+/-};Dmrta2^{+/-}$ (Konno et al., 2019), or $Dmrt3^{-/-};Dmrta2^{-/-}$ compared to $Dmrt3^{+/+};Dmrta2^{+/+}$ (Desmaris et al., 2018)]. The genes upregulated in *Dmrt3* and *Dmrta2* double-knockout mice significantly overlapped with the DEGs upregulated during neuronal differentiation in our RNA-seq analysis (Fig. 7D,E, Table S1). In contrast, the expression levels of downregulated genes in the double knockout mice decreased during neuronal differentiation (Fig. 7D,E). Taken together, these results suggested that Dmrt3 and Dmrta2 maintain the closed chromatin state of neuronal genes in NPCs and that their reduction facilitates chromatin opening during neuronal differentiation.

## DISCUSSION

The competence of neurons for their functions, such as neuronal subtype-specific functions and responsiveness to external stimuli, is established during neuronal differentiation of NPCs. Although several studies have examined changes using biochemical analysis of *in vitro* primary cultures of neurons, the *in vitro* neuronal differentiation process differs considerably from the *in vivo* process, which involves neuronal migration and drastic morphological changes. As the brain, even at the embryonic stage, consists of many cell types, including different subtypes of neurons, glial cells and non-neural cells, it is necessary to label and isolate specific lineages of neurons to analyze the neuronal differentiation process *in vivo*. Telley et al. reported a FlashTag method to label cortical neurons with the same birthdate via random conjugation of a fluorescent dye to intracellular proteins (Telley et al., 2016). FlashTag is useful because of its simplicity; however, the proteins labeled with fluorescence degrade, thus shortening the possible tracing duration time. Using Neurog2-CreERT2 transgenic mice, Hirata et al. succeeded in genetically and permanently labeling cortical neurons with the same birthdates (Hirata et al., 2016, 2021; Hou et al., 2019). Our novel methods of introducing pNeuroD1-ERT2CreERT2 via *in utero* electroporation have advantages when using wild-type mice and permanently labeling cortical neurons with the same birthdate.

Leveraging these advantages, we traced the differentiation of neurons produced from NPCs around E12 for 4 days and isolated them to perform RNA-seq and DNase-seq analyses simultaneously. Seventy-five percent of the DEGs (1742 of 2318 genes) observed by Telley et al. were also annotated as DEGs in our analysis, indicating the feasibility of our new labeling method for labeling newborn neurons at similar stages (Telley et al., 2016). In addition, our data revealed that the transcription levels of more than half of the protein-coding genes changed significantly during neuronal differentiation, indicating global changes in transcriptional states during neuronal differentiation *in vivo*, as observed in *in vitro* primary cultures (Kaur et al., 2014).

Chromatin openness, as examined using DNase-seq, plays an important role in gene transcription. In our analysis, there were more opening regions at promoter regions during neuronal differentiation than closed regions, and the changes at the promoter, especially at E16.0, correlated with changes in transcription levels of nearby genes, indicating the contribution of chromatin regulation to changes in transcription patterns. This differs from the results of the ATAC-seq of granule neurons in the dentate gyrus with electroconvulsive stimulation, which showed that most changes in chromatin openness were observed in the intronic and distal regions (Su et al., 2017). Therefore, chromatin regulation may differ during developmental and transient changes in gene transcription. Previous reports have shown that chromatin regulation at the enhancer site also contributes to regulation of the transcriptional state during development (Heintzman et al., 2009; Song et al., 2011). Notably, 79% of the regions that became accessible from E12.0 to E16.0 were located in non-promoter regions (Fig. 3E); 19.7% of these non-promoter opening regions were annotated as enhancer regions based on the EnhancerAtlas 2.0 database. Future studies focusing on other chromatin regulators, such as epigenetic modifications and three-dimensional chromatin structures, will help elucidate the role of enhancer region opening during early neuronal differentiation.

In addition to transcriptional activity, chromatin opening contributes to the preparatory state for future activation. Our results suggest that chromatin opening without activation of the gene loci is a feature of deep-layer excitatory neurons, such as that observed in deep layer-specific genes and BDNF stimulus-dependent genes, during embryonic differentiation. Interestingly, genes closing during the NPC to E16 neuron transition were enriched in the upper layer-specific genes. These results suggest that chromatin regulation during the embryonic stage confers layer-specific neuronal functions. In the case of genes responsive to external stimuli, previous reports have shown that electroconvulsive stimulation of granule neurons in the dentate gyrus induces chromatin opening, mainly in the distal regulatory elements of responsive genes, in a Fos-dependent manner (Su et al., 2017). Combined with our results showing the enrichment of Fos and the AP-1 complex containing Fos in the opening regions, responsive genes may acquire a preparatory state by opening their promoter regions at the embryonic stage and activating their transcription through their enhancer regions.

To identify chromatin regulators during neuronal differentiation, we used the ChIP-Atlas database of the binding sites of TFs determined using ChIP-seq experiments (Oki et al., 2018). We found that the binding sites for histone methyltransferases, such as Brd4 and Ash2l for H3K4me3 and components of the PcG for H3K27me3, which are involved in creating bivalent marks determined in ESCs, were specifically enriched in DHSs identified in E16 neurons. Our ChIP-seq analyses of H3K4me3 and H3K27me3 marks in NPCs showed that bivalent genes in NPCs became active during neuronal differentiation from NPCs to E16 neurons. This suggests that opening of bivalent genes contributes to neuronal gene activation. Previous studies have proposed several mechanisms for the activation of bivalent genes, such as the contribution of H3K27me3 demethylase and the chromatin remodeling complex included in the PcG or TrxG complexes (Dhar et al., 2016; Narayanan et al., 2015; Tang et al., 2020). We examined changes in the expression of genes encoding the components of PcG or TrxG in our RNA-seq dataset and found that more than half of them showed significant alterations during neuronal differentiation (Table S8), suggesting their potential contribution to chromatin remodeling during this process. In addition, our motif analysis of the open and bivalent regions during differentiation from E12.0 NPCs to E16.0 neurons using HOMER software revealed enrichment of the DNA sequence, which is similar to the binding motif of Dmrt3. Previous reports have shown that Dmrt3 and Dmrta2 (also known as Dmrt5) are important for maintaining the undifferentiated state of NPCs by repressing neurogenic genes (Desmaris et al., 2018; Konno et al., 2012, 2019). To investigate whether Dmrt3 and Dmrta2 regulate the opening of genes, we analyzed the ChIP-seq results for Dmrt3 or Dmrta2 and the RNA-seq data of NPCs from the brains of *Dmrt3* and *Dmrta2* double-knockout mice. We found that genes bound by Dmrt3 or Dmrta2 were enriched in opening genes, and that Dmrt3- and Dmrta2-repressed genes were upregulated during neuronal differentiation. Considering that Dmrt3 and Dmrta2 act as transcriptional repressors, these results suggest that Dmrt3 and Dmrta2 play a crucial role in maintaining the closed chromatin state of opening genes in NPCs and are important for regulating the timing of differentiation in NPCs.

In this study, we describe transcriptomic and chromatin changes during the differentiation of deep-layer neurons in the mouse neocortex using a novel labeling method for a specific lineage of neurons and the opening of the chromatin state in preparation for future gene activation. However, we focused only on deep-layer neurons, which is a limitation of the study. Future experiments that include similar analyses of upper-layer neurons and non-neuronal cells will provide a more comprehensive understanding of the role of chromatin accessibility in cerebral cortex development.

## MATERIALS AND METHODS

### Ethics statement

All animals were maintained and studied according to the protocols approved by the Animal Care and Use Committee of the University of Tokyo (approval numbers: P25-8 and P30-4 at the Graduate School of Pharmaceutical Sciences and 0421 and A2022IQB001-06 at the Institute for Quantitative Biosciences). All procedures were performed in accordance with The University of Tokyo guidelines for the care and use of laboratory animals and the ARRIVE guidelines.

### Mouse maintenance and preparation of pregnant mice

JCL:ICR (CLEA Japan), Slc:ICR (Japan SLC) and C57BL/6J mice were housed in a temperature- and humidity-controlled environment (23±3°C and 50±15%, respectively) under a 12-h light/dark cycle. The animals were housed in sterile cages (Innocage, Innovive) containing bedding chips (PALSOFT, Oriental Yeast), with two to six mice per cage, and provided with irradiated food (CE-2, CLEA Japan) and filtered water *ad libitum*. For *in utero* electroporation experiments, mating was limited to a 2-h period, and the presence of a vaginal plug was used to confirm the onset of gestation (E0.0).

### Plasmid constructs

pAAV-DIO-GFP (pAAV-Ef1a-DIO-EGFP-WPRE-pA) and pNeuroD1-IRES-GFP were provided by Frank Polleux (Jossin and Cooper, 2011). To construct pNeuroD1-ERT2CreERT2 (Addgene plasmid #237402), ERT2CreERT2 was digested with pCAG-ERT2CreERT2 (Addgene plasmid #13777) and cloned into pNeuroD1-IRES-GFP, which was digested with the same restriction enzymes. The pCAG-mCherry vector was constructed as described previously (Maeda et al., 2024).

### *In utero* electroporation

E12.0 mice were administered a sodium pentobarbital-based anesthetic and the uterine horn was exposed. The embryo was pinched between a flexible fiber-optic cable and a finger, and plasmid DNA (pNeuroD1-ERT2CreERT2, 0.5 µg µl⁻¹; pAAV-DIO-GFP, 1 µg l⁻¹; pNeuroD1-IRES-GFP, 0.5 µg µl⁻¹; pCAG-mCherry, 1 µg l⁻¹) containing 0.01% Fast Green was injected into the lateral ventricle. The uterus was held between tweezer-type electrodes (CUY650P3, Nepa Gene) and electroporated with the CUY21EDIT or NEPA21 system (Nepa Gene) according to the following program: four pulses of 30 V with a 50-ms pulse width and 950-ms pulse interval. After the injection and electroporation, the uterine horn was returned to its original location. The mice recovered on a heating plate and were maintained at 38°C until they regained consciousness following anesthesia.

### Immunohistochemistry

Immunofluorescence staining was performed as described previously (Eto et al., 2020). E13.0 embryos were directly fixed with 4% paraformaldehyde (PFA) in PBS at 4°C for 2-3 h. E14.0 and E16.0 embryos were perfused with 4% PFA in PBS, and the isolated brains were postfixed with 4% PFA in PBS at 4°C for 2-3 h. After equilibration with 30% (w/v) sucrose in PBS, the fixed brains were embedded in OCT compound (Tissue-Tek) and frozen. Coronal cryosections (~15 µm) on slide glasses were exposed to TBS with 0.1% Triton X-100 (TBS-T) and 3% bovine serum albumin (blocking buffer) and incubated with anti-GFP (Nacalai, GF090R; 1:1000) and anti-DsRed (for mCherry; MBL, PM005; 1:000) antibodies in blocking buffer overnight at 4°C. After washing with TBS-T, the slides were incubated with Alexa Fluor-conjugated secondary antibodies (Thermo Fisher Scientific, A32794 and A21208; 1:1000) and Hoechst 33342 (Thermo Fisher Scientific, H3570; 1:10000) in blocking buffer and mounted in Mowiol (Calbiochem). All fluorescence images were obtained using a laser confocal microscope (Leica TCS-SP5) and analyzed using ImageJ software (NIH).

### Isolation of NPCs and GFP-labeled neurons using FACS

NPCs and neurons were isolated using a previously established protocol (Kishi and Gotoh, 2021). After electroporation, the neocortices of non-electroporated E12.0, E13.0, E14.0 and E16.0 embryos were dissected and subjected to enzymatic digestion using neuron dissociation solution (FujiFilm Wako Chemicals). For E12.0 NPCs, the dissociated cells were stained with an allophycocyanin-conjugated anti-CD133 antibody (BioLegend, 141208; 1:400). E12.0 NPCs (CD133-high cells) or E13.0, E14.0 and E16.0 neurons (GFP-positive cells) were isolated using a FACSAria instrument (Becton Biosciences).

### RT-qPCR

RT-qPCR was performed as described previously (Kishi and Gotoh, 2021). Total RNA was extracted from isolated cells using FACS with RNAiso Plus (Takara Bio) and subjected to reverse transcription using ReverTra Ace qPCR RT master mix with gDNA remover (Toyobo). The concentrations of the target cDNA were determined using the THUNDERBIRD SYBR qPCR Mix (Toyobo) on a LightCycler 480 or LC96 instrument (Roche). Expression levels were calculated after absolute quantification and normalized to levels of the *Actb* housekeeping gene. The primer sequences were as follows: *Actb*, AATAGTCATTCCAAGTATCCATGAAA and GCGACCATCCTCCTCTTAG; *Nes*, TGAAGCACTGGGAAGAGTA and TAACTCATCTGCCTCACTGTC; *NeuroD1*, TACGACATGAACGGCT-GCTA and TCTCCACCTTGCTCACTTT; *Tubb3*, ACACAGACGA-GACCTACT and GCAGACACAAGGTGGTT; *Gabrb2*, ATGCCATCAA-TTCTGATTACCA and TAATTCCTAATGCAACCCGTG.

### RNA-seq

Libraries for RNA-seq analysis were constructed from total RNA isolated in two independent experiments at each stage of RT-qPCR analysis. A TruSeq RNA sample preparation kit (Illumina) was used for template preparation, followed by sequencing on a HiSeq platform (Illumina) to obtain 100-base paired-end reads. More than 20 million fragments were analyzed. For layer-specific gene expression analysis, raw sequence files were obtained from the Gene Expression Omnibus database (GSE27243; GSM673634 for layers 1-3 and GSM673639 for layer 6) (Belgard et al., 2011). For gene expression analysis of *Dmrta3* and *Dmrta2* double-knockout brains, raw sequence files were obtained from the DNA Data Bank of the Japan (DDBJ) or Gene Expression Omnibus databases (DRA004335 or GSE108611) (Desmaris et al., 2018; Konno et al., 2019). Adaptor sequences, pNeuroD1 plasmid-containing sequences, low-quality sequences (MAPQ<30) and poly-A sequences were removed using Trimmomatic, Bowtie, Fastx toolkit and PRINSEQ software, respectively (Bolger et al., 2014; Langmead et al., 2009; Schmieder and Edwards, 2011). Reads were mapped to the reference mouse genome (mm10) using HISAT2 (Kim et al., 2019). Reads mapped to ribosomal RNA genes and sex chromosomes were removed using SAMtools and BEDTools (Li et al., 2009; Quinlan and Hall, 2010) as pooled cells from both male and female embryos were used in this study. Reads mapped to protein-coding genes based on the GENCODE v.M20 mouse genome reference (Frankish et al., 2018) were counted using featureCounts and normalized to iDEGS/edgeR (Liao et al., 2014; Robinson et al., 2010). To compare the expression levels between different genes, the values were normalized by the gene length, which were called 'expression value' in this study. DEGs were determined as genes with FDR<0.05, calculated using iDEGES/edgeR. Due to the availability of only a single replicate of RNA-seq data in the Konno et al. (2019) study, we defined DEGs as those exhibiting a greater than twofold change in expression levels. GO analysis was performed using DAVID software (Huang et al., 2009a,b). GSEA was performed using software provided by the Broad Institute (Mootha et al., 2003; Subramanian et al., 2005).

To display data accurately for genes with an expression level of zero on a logarithmic scale graph, all dot plots of expression levels are presented as 'expression value +1'. Data with values greater than the maximum are not shown as outliers in any of the dot plot graphs of expression levels. All raw data are provided in Table S1.

### DNase-seq

Cells isolated from two independent FACS experiments were resuspended in nuclear buffer A [85 mM KCl, 5.5% sucrose, 10 mM Tris-HCl (pH 7.5), 0.5 mM spermidine and 0.2 mM EDTA] and gently mixed with the same volume of nuclear buffer B (nuclear buffer A with 0.1% NP-40). After centrifuging at 600 *g* for 5 min at 4°C and removing the supernatant, the pelleted nuclei were resuspended in nuclear buffer R [85 mM KCl, 5.5% sucrose, 10 mM Tris-HCl (pH 7.5), 3 mM MgCl₂ and 1.5 mM CaCl₂], and

DNase I (Takara Bio) was added to a final concentration of 2 U ml$^{-1}$ and incubated for 30 min at 37°C. After quenching the reaction by adding lysis buffer [1% sodium dodecyl sulfate (SDS), 50 mM Tris-HCl (pH 8.0) and 10 mM EDTA], the digested DNA was purified using phenol-chloroform-isoamyl alcohol and subjected to agarose gel electrophoresis. DNA less than 1 kb in length was purified using a FastGene gel/PCR extraction kit (NIPPON Genetics).

Libraries for DNase-seq analysis were constructed using a TruSeq ChIP sample preparation kit (Illumina), which was used for template preparation, followed by sequencing on a HiSeq platform (Illumina) to obtain 36-base paired-end reads. More than 20 million fragments were analyzed. pNeuroD1 and pAAV-DIO-EGFP plasmid-containing sequences were removed using Bowtie software. Reads were mapped to the reference mouse genome (mm10) using bowties, allowing a maximum of two mismatches. Only uniquely aligned reads were used for further analyses. Sequenced reads mapped to known ENCODE blacklist regions (Amemiya et al., 2019; ENCODE Project Consortium et al., 2012) and sex chromosomes were removed using BEDTools and SAMtools. The peaks were determined by F-Seq as regions with $P<10^{-6}$, and the peaks detected in both replicates were determined as DHSs. The opening or closing regions were defined as DHSs that appeared or disappeared during differentiation, respectively.

To determine genomic features, protein-coding genes annotated in GENCODE v.M20 were used. Promoters were determined to be 1 kb upstream and 0.5 kb downstream of the transcription start sites, and exons and introns did not include promoters. Downstream sequences were determined to be 1 kb downstream of the transcription termination site without promoters, exons, or introns. The distal regions were identified as regions that have not yet been characterized. Enhancers were determined as an integrated list of enhancers of NPC and NeuronCortical in the EnhancerAtlas 2.0 (Gao and Qian, 2019a). Briefly, the EnhancerAtlas 2.0 database identified enhancer regions by integrating public data from the following: (1) epigenetic modifications, using ChIP-seq for histone modifications and TFs specific to the target cell type; (2) chromatin accessibility, assessed by DNase-seq, formaldehyde-assisted isolation of regulatory elements (FAIRE-seq) and ATAC-seq; and (3) eRNA expression, analyzed using Pol II ChIP-seq, genomic run-on (GRO-seq), and cap analysis of gene expression (CAGE). Enhancer-gene pairs were determined using EAGLE software, which integrates correlations between enhancer activity, gene expression levels, and the distance between enhancers and genes (Gao and Qian, 2019b). ChIP-Atlas was utilized to identify the TFs that especially bind to DHSs determined in E16.0 neurons (Oki et al., 2018). IGV software (Broad Institute) was utilized for visualizing DNase-seq signals.

## Microarray

The cortex from E15 C57BL/6J mice was isolated from three independent experiments, dissociated with neuronal dissociation solutions (FujiFilm Wako Chemicals), and plated at a density of approximately 1.2 cells/cm$^2$ on poly-D-lysine-coated dishes in neurobasal medium (Gibco) supplemented with 1% Glutamax (Gibco) and 2% B27 (Gibco). After 6 days, the culture was treated with 50 ng ml$^{-1}$ BDNF (Sigma-Aldrich) for 1 h. RNA isolated for RT-qPCR was analyzed using a SurePrint G3 mouse GE 8×60 K microarray (Agilent), according to the manufacturer's instructions (Agilent SureScan). The data were normalized by percentile shift (75%) using GeneSpring software. DEGs were identified using unpaired and two-tailed Student's $t$-test as genes with $q<0.05$.

## ChIP-seq

ChIP was performed as described previously (Eto et al., 2020). NPCs isolated from two independent experiments using FACS from E12.0 embryos were fixed with 1% formaldehyde and stored at −80°C until further use. The cells were thawed, suspended in radioimmunoprecipitation assay (RIPA) buffer for sonication (10 mM Tris-HCl at pH 8.0, 1 mM EDTA, 140 mM NaCl, 1% Triton X-100, 0.1% SDS and 0.1% sodium deoxycholate), and subjected to ultrasonic treatment using a Picoruptor (15 cycles of 30 s ON and 30 s OFF) (Diagenode). The cell lysates were then diluted with RIPA buffer for immunoprecipitation (50 mM Tris-HCl at pH 8.0, 150 mM NaCl, 2 mM EDTA, 1% Nonidet P 40, 0.1% SDS and 0.5% sodium deoxycholate), incubated for 1 h at 4°C with protein A/G magnetic

beads (Pierce) to clear non-specific binding, and then incubated overnight at 4°C with protein A/G magnetic beads that had previously been incubated overnight at 4°C with 2 μl of antibodies against H3K27me3 (MBL, MABI0323; 2 μg/sample) or H3K4me3 (MBL, MABI0304; 2 μg/sample). The beads were isolated and washed thrice with wash buffer (2 mM EDTA, 150 mM NaCl, 0.1% SDS, 1% Triton X-100 and 20 mM Tris HCl, pH 8.0) and then once with wash buffer containing 500 mM NaCl. Immune complexes were eluted from the beads for 15 min at 65°C in direct elution buffer (10 mM Tris HCl at pH 8.0, 5 mM EDTA, 300 mM NaCl and 0.5% SDS) and then subjected to digestion with proteinase K (Nacalai Tesque) for >6 h at 37°C; they were decrosslinked by incubating at 65°C for >6 h. Immunoprecipitated DNA was purified using phenol-chloroform-isoamyl alcohol.

Libraries for ChIP-seq analysis were constructed using the TruSeq ChIP sample preparation kit (Illumina), which was used for template preparation, followed by sequencing on a HiSeq platform (Illumina) to obtain 36-base paired-end reads. More than 20 million fragments were analyzed. For ChIP-seq of Dmrta3 and Dmrta2, raw sequence files were obtained from the DDBJ database (DRA004335) (Konno et al., 2019). Processing, mapping, and peak calling were performed as described for DNase-seq. The peaks were identified using F-Seq software (Boyle et al., 2008a).

## Motif enrichment analysis

Motif enrichment analysis was performed using HOMER against background sequences randomly selected from the whole genome with similar GC percentage (except for repeat sequences) (Heinz et al., 2010).

## Statistical analysis

Data were compared using analysis of variance (ANOVA), followed by Tukey's multiple comparison test, Friedman's test and Dunn's multiple comparison test. $P$-values <0.05 were considered statistically significant. Fisher's exact test was performed to test for overlap between the two gene sets.

### Acknowledgements
We thank F. Polleux (Columbia University) for providing the plasmid pNeuroD1-IRES-GFP; N. Shinoda, M. Miura (University of Tokyo) and Y. Yamaguchi (Hokkaido University) for helping with microarray analysis; R. Nakato, K. Kadota, T. Horiuchi (University of Tokyo) and S. Oki (Kumamoto University) for advice regarding sequence analysis; the Human Genome Center (University of Tokyo) for providing the super-computing resource; the One-Stop Sharing Facility Center for Future Drug Discoveries (University of Tokyo) for providing the FACS facility; R. Nagayoshi and Y. Kakeya (University of Tokyo) for technical assistance; and members of the Gotoh and Kishi laboratories for helpful discussion.

### Competing interests
The authors declare no competing or financial interests.

### Author contributions
Conceptualization: S.S., Y.G., Y.K.; Data curation: S.S., M.S., Y.S., Y.K.; Formal analysis: S.S., Y.K.; Funding acquisition: S.S., Y.G., Y.K.; Investigation: S.S., Y.M., M.S., D.K., K.K., F.M., Y.S., Y.K.; Methodology: Y.M., Y.K.; Project administration: Y.G., Y.K.; Supervision: Y.G., Y.K.; Validation: Y.K.; Visualization: S.S., Y.K.; Writing – original draft: S.S., Y.K.; Writing – review & editing: S.S., Y.G., Y.K.

### Funding
This research was supported by the Japan Agency for Medical Research and Development via the AMED-CREST (JP23gm1310004 to Y.G.) and AMED-PRIME (JP22gm6110021 to Y.K.) programs; the Ministry of Education, Culture, Sports, Science and Technology (MEXT)/Japan Society for the Promotion of Science (JSPS) KAKENHI (JP22H00431 to Y.G.; 16H06279, JP22H04687, 23H04214 and 24H01227 to Y.K.); the Takeda Science Foundation; the Uehara Memorial Foundation; the Asahi Glass Foundation; the Chugai Foundation for Innovative Drug Discovery Science; the Astellas Foundation for Research on Metabolic Disorders; the Naito Foundation; the SECOM Science and Technology Foundation; the Ono Pharmaceutical Foundation for Oncology, Immunology, and Neurology; and Kurata Grants from The Hitachi Global Foundation. Open Access funding provided by the University of Tokyo. Deposited in PMC for immediate release.

### Data and resource availability
Sequence and microarray data were deposited in the DDBJ Sequence Read Archive under the following accession codes: DRA015915 (RNA-seq), DRA015284,

DRA018693 (DNase-seq) and DRA018694 (H3K4me3 and H3K27me3 ChIP-seq). Processed sequences and microarray data were deposited in the DDBJ Gene Expression Archive under accession codes E-GEAD-803 (microarray of BDNF stimulation), E-GEAD-859 (peak files of DHSs) and E-GEAD-860 (BigWig files of ChIP-seq).

## Peer review history

The peer review history is available online at https://journals.biologists.com/dev/lookup/doi/10.1242/dev.204564.reviewer-comments.pdf

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
