## [Peer Review File · Development (Cambridge, England)]

***In vivo* transition in chromatin accessibility during differentiation of deep-layer excitatory neurons in the neocortex**

Seishin Sakai, Yurie Maeda, Mai Saeki, Daijiro Konno, Keita Kawaji, Fumio Matsuzaki, Yutaka Suzuki, Yukiko Gotoh and Yusuke Kishi
DOI: 10.1242/dev.204564

Editor: James Briscoe

Review timeline

Submission to Review Commons:	9 February 2024
Submission to Development:	26 November 2024
Editorial decision:	6 December 2024
First revision received:	24 December 2024
Editorial decision:	30 December 2024
Second revision received:	16 April 2025
Editorial decision:	5 May 2025
Third revision received:	25 May 2025
Accepted:	27 May 2025

Reviewer 1

Evidence, reproducibility and clarity

The authors have developed a method for labeling a specific stage of differentiating neurons. Using this approach, they tracked the four-day differentiation process of deep-layer excitatory neurons in the mouse embryonic cortex. They investigated genome-wide changes in transcription patterns and chromatin accessibility using RNA-seq and DNase-seq. Additionally, they provided H3K4me3 and H3K27me3 ChIP-seq data from E12.0 NPCs. This resulting omics data would be a valuable resource for the field. While initial data analyses show potentially interesting findings, only part of the analyses are presented in the figures, lacking sufficient detail. Before publishing the manuscript, the authors should include more comprehensive analyses of their datasets. Specific suggestions are below.

In Figure 1c, the actual values of the differentially expressed genes are unclear. Is this a Z-score? Please provide the log₂ expression values and specify the scale used for the heatmap and clustering.

Figure 4 focuses on promoter-specific chromatin accessibility analysis. The author can process the data similarly to the transcription data. They should identify differentially accessible promoter regions across E13.0 to E16.0 and generate a heatmap with clustering. Additionally, the author should provide matched gene expression data, either in the form of a heatmap or box plot, corresponding to those differentially accessible promoter regions. Currently, Figure 4 only presents E16.0 data compared to E12.0, which is not comprehensive.

Figure 5: It is somewhat unusual that the authors used microarray instead of RNA-seq for the BDNA stimulation of in vitro cortical neurons. Please provide a justification for this choice.

Figure 6: again, the data analyses are not comprehensively presented. What are the gene expression profiles of the other clusters (H3K27me₃⁺, H3K4me₃-/H3K27me₃⁻, H3K4me₃⁺)? Additionally, the sequencing data is inaccessible, and it is unclear how many samples (e.g., replicates) were used in this study for RNA-seq, DNase-seq, and ChIP-seq.

Significance

Multi-omics data from the differentiation process of deep-layer excitatory neurons would be a valuable resource for the field.

Reviewer 2

Evidence, reproducibility and clarity

Summary:

The manuscript from Sakai et al. examines changes in chromatin accessibility during the differentiation of deep-layer excitatory neurons in the neocortex. The authors establish a novel genetic labelling method that tracks differentiating neurons based on their birthdates allowing following neuronal differentiation in vivo. By combining RNA-seq and DNase-seq they provide a comprehensive dataset of gene expression and chromatin accessibility changes during neuronal differentiation of deep-layer neurons and reveal that key genes linked to mature neuronal functions and bivalent genes in neural precursor cells become accessible during early differentiation. These findings underscore the crucial role of chromatin regulation in preparing neurons for maturation and unravel novel key insights into the regulatory mechanisms governing deep-layer neuronal differentiation.

Overall, this manuscript presents a novel technique for tracking neuron development from NPCs with specific birthdates. However, in its current form, it is largely descriptive and relies on correlative observations rather than elucidating a clear mechanism underlying chromatin and transcriptional changes. The provided data could be further leveraged to gain deeper insights into the molecular mechanisms governing deep-layer neuron development.

One additional point, which may be beyond the scope of this paper, is that to demonstrate the temporal resolution of this birthdate tracking method robustly, the authors should also apply the technique to upper-layer neuron development and compare developmental differences that were previously challenging to capture due to lower resolution.

Major comments:

The authors have generated extensive RNA- and DNase-seq datasets across different developmental time points following birthdate labelling. However, the bioinformatics analyses and interpretations are limited and need further clarification and refinement:

1. The violin plots used to demonstrate expression and accessibility changes across developmental time points and the conclusions drawn from them are not convincing. The authors used a rank test to assess significant changes in expression, which only indicates the enrichment of genes with increased or decreased expression in each group. This cannot be directly interpreted as "significant upregulation." For instance, in Figures 4a and 4b, similar violin plots yield different statistical outcomes. The mean values on both graphs are comparable, yet Figure 4a suggests significant changes, while Figure 4b does not conclude significant downregulation of closing DHS genes. This is unconvincing. A more robust approach would be identifying DEGs between time points and analysing functional terms associated with these genes. The current plots do not support interpretations of gene upregulation, as each dot represents a gene, and the violin plot serves more as a population representation. The authors should either revisit their explanations and conclusions or include additional analyses and appropriate plots that support their claims of significant upregulation and downregulation of specific genes during development.
2. Figure 6b lacks clarity regarding the cutoff value used to categorise genes as K4me3 and K27me3 negative or positive from the heatmap. Even the "K4me3 negative" cluster displays a detectable signal of the mark, albeit at lower levels. Since only one plot of the entire gene body is provided, it is unclear what levels of enrichment are present, particularly at the promoter region. The authors are encouraged to provide additional informative plots and analyses of this ChIP-seq experiment, as this is a critical point where they draw conclusions about bivalent genes. This

would not only strengthen their claims but could also uncover additional findings with more detailed analyses. A heatmap of clustered ChIP-seq signals of K4me3 and K27me3 alongside expression levels of the same genes (similar to Figure 2c) and differential accessibility (e.g., between NPC and E16) would better visualise and correlate histone modifications with chromatin and gene expression states.

3. The DNase-seq dataset can be better utilised to investigate differentially accessible motifs through development. Is this something the authors already looked into? This could strengthen mechanism investigation together with the ChIP-atlas results in Fig.6a

4. The two distinct modes of H3K4me3 enrichment observed are not addressed and should be explained. Which genes belong to these two clusters? Is there a difference in DHS and gene expression between them?

5. The same concern regarding the use of violin plots to correlate gene expression with bivalent genes through development (Figure 6c) as mentioned earlier. It would be better to use DEGs and intersect them. This is particularly important given the wide range of gene expression levels in the already poised state.

6. The authors limited their analyses to promoter/gene body regions. A survey of the bivalent marks and accessibility at enhancer regions would be also beneficial for understanding the changes at the chromatin landscape through development.

7. The authors begin by examining TFs enriched at E16 DHS regions and suggest that TrxG and PcG factors are highly enriched in neurons, initiating their investigation of bivalent marks. However, they later conclude that bivalent marks are present in the NPC state and later become accessible. It is unclear why PRC factors would be enriched at the neuronal stage when the authors conclude that the chromatin becomes more open (potentially by removal of K27me3). The authors should refine this section of the manuscript to better rationalise their methodology and results.

8. Do the authors find any expressional changes of the suggested candidate proteins at the RNA or protein levels through development?

9. The mechanisms driving the activation and expression of poised neuronal genes through the development of deep-layer neurons is not uncovered. The authors suggest certain histone modifiers and the DNA methyltransferase Dnmt3 as potential drivers of chromatin landscape and transcriptional regulation changes; however, this remains speculative, as there is no direct evidence or validation of these factors binding to the identified target regions or changes in DNA methylation states. The authors should provide validation of their candidate factors' presence at potential targets, as well as changes in DNA methylation if they want to conclude these as the mechanisms driving deep-layer neuron development.

Minor comments:

1. The manuscript would improve with proofreading by a native English speaker.
2. The motif analysis can be included in the main figures.

Significance

By introducing a novel genetic labelling method that tracks neurons based on their birthdates, the study provides a precise way to examine differentiation in vivo, adding valuable insights beyond traditional in vitro approaches. The combination of RNA-seq and DNase-seq analyses reveals how chromatin accessibility changes, particularly in bivalent genes, play a crucial role in neuronal maturation. This work highlights the importance of chromatin dynamics in establishing neuronal identity. The techniques and findings provide a useful framework for future studies, offering a path for deeper exploration of chromatin regulation across different neuronal types, stages of development, or disease contexts, making it a valuable contribution to the field of developmental neurobiology.

While the manuscript suggests the involvement of chromatin regulators such as Trithorax and Polycomb proteins, as well as Dnmt3 and DNA methylation, it lacks direct mechanistic evidence, such as ChIP-seq, bisulfite-seq, or loss-of-function experiments, to substantiate these claims.

The study focuses exclusively on deep-layer excitatory neurons, without comparisons to other neuronal subtypes or non-neuronal cells. Including such comparisons would help determine whether the observed chromatin changes are unique to this specific population or part of a

broader developmental process.

The bioinformatics analyses and interpretations are limited and require further clarification and refinement.

The proposed mechanisms are not fully explored, leaving the manuscript largely descriptive rather than providing a detailed mechanistic understanding.

Reviewer 3

Evidence, reproducibility and clarity

In this manuscript the authors use in utero electroporation of tamoxifen inducible reporters to permanently mark cortical neurons with a common birthdate. They then FACS harvest these cells for bulk DNase seq and RNA seq to see changes in chromatin regulation and gene expression as these newborn immature cortical neurons become deep layer neurons. As has been shown in prior studies that have addressed other neuronal types or used different methods to isolate developmental cell stages in the CNS, the authors find correlated changes between the opening or closing of chromatin with changes in gene expression.

They use this information to localize chromatin marks that are associated with the differential expression of genes and conclude that many of the differential genes are bivalent for active and repressive chromatin marks. Finally the authors cross this dataset with a microarray they did of BDNF-inducible genes in cortical culture and suggest enrichment of this program in the differentially regulated gene set from in vivo.

Significance

The idea that chromatin regulation coordinates developmental changes in gene expression in neurons has been addressed with several different strategies over the past decade including prior strategies that allow for isolation of neurons with common birth dates. Many current strategies (well cited by the authors) use single cell sequencing and computational algorithms to deconvolve differentiation state from complex mixtures. This study takes an alternative approach to experimentally label these developmental stages which is nice to see for the validation of ground truth. However the study does not go far beyond current knowledge to use this method to add new concepts to the field. The main point of innovation seems to be the observation that the newborn neurons are primed at the chromatin level to express deep layer markers at the time they are born during embryonic life. This is useful to see but not unexpected on the basis of large scale single cell datasets. They also show that bivalent promoters prime developmental stage specific gene expression (in addition to the well-established function of this form of regulation in fate determination), however this too has been shown already in other neuron types.

In addition to these conceptual limitations, there are some poorly supported comments in the text. For example, the fact that their microarray shows some genes in a category called "apoptosis" that are BDNF-sensitive does not meaningful suggest that BDNF induces excitotoxicity in embryonic cortical culture. BDNF has been well established as a survival factor for many kinds of neurons and is a common additive to serum-free media supplements (like B27). The appearance of "apoptosis" terms in the upregulated genes on the microarray more likely suggests either that the microarray is a poor detector of differential gene expression or that the genes in question are inaccurately categorized as "apoptotic" (GO terms are not terribly specific indicators of gene function). If the authors really wanted to test if BDNF was inducing apoptosis their cultures they could test this. However to use only the GO term data in such a strong statement about the biology of their system caused me to question the rigor of either their data or their analysis.

A second example is the section about promoters being the focus of their discussion for DHS sites. Sure figure 3c shows promoters are more likely to be open compared with their contribution to the genome overall, but this is entirely expected since they are major gene TF binding sites, which is what DNase detects. However promoters do not look to be more likely to be differentially regulated over time (3c vs 3e), and the statement that promoters are more enriched in opening compared with closing sites would require a statistical statement. Distal DHS sites appear equally more abundant in opening sites too.

Author response to reviewers' comments

1. General Statements [optional]

We are grateful to the reviewers for their many valuable suggestions for improving this paper. In particular, we fully understand the points raised by Reviewers #1 and #2 regarding the insufficient data analysis and the points raised by Reviewers #2 and #3 regarding the insufficient analysis of the mechanism. In future revisions, we will perform sufficient analysis of our datasets and we will also conduct an analysis focusing on *Dmrt3* to investigate the mechanisms for chromatin accessibility and changes in gene expression during neuronal differentiation. We will also make revisions to address other minor points.

2. Description of the planned revisions

Reviewer #1 (Evidence, reproducibility and clarity (Required)):

The authors have developed a method for labeling a specific stage of differentiating neurons. Using this approach, they tracked the four-day differentiation process of deep-layer excitatory neurons in the mouse embryonic cortex. They investigated genome-wide changes in transcription patterns and chromatin accessibility using RNA-seq and DNase-seq. Additionally, they provided H3K4me3 and H3K27me3 ChIP-seq data from E12.0 NPCs. This resulting omics data would be a valuable resource for the field. While initial data analyses show potentially interesting findings, only part of the analyses are presented in the figures, lacking sufficient detail. Before publishing the manuscript, the authors should include more comprehensive analyses of their datasets. Specific suggestions are below.

We appreciate this reviewer's positive comments describing our study as 'a valuable resource for the field.' We plan to revise the paper, as noted below, to address this reviewer's concerns.

Figure 4 focuses on promoter-specific chromatin accessibility analysis. The author can process the data similarly to the transcription data. They should identify differentially accessible promoter regions across E13.0 to E16.0 and generate a heatmap with clustering. Additionally, the author should provide matched gene expression data, either in the form of a heatmap or box plot, corresponding to those differentially accessible promoter regions. Currently, Figure 4 only presents E16.0 data compared to E12.0, which is not comprehensive.

We thank the reviewer for the useful suggestions. In the following submission, we will determine gene sets for all chromatin accessibility change patterns, not just open/closed gene sets from E12 to E16. We will then illustrate the changes in gene expression for each gene set.

Reviewer #1 (Significance (Required)):

Multi-omics data from the differentiation process of deep-layer excitatory neurons would be a valuable resource for the field.

Once again, we would like to thank the reviewers for their positive comments.

Reviewer #2 (Evidence, reproducibility and clarity (Required)):

Summary:

The manuscript from Sakai et al. examines changes in chromatin accessibility during the differentiation of deep-layer excitatory neurons in the neocortex. The authors establish a novel

genetic labelling method that tracks differentiating neurons based on their birthdates allowing following neuronal differentiation *in vivo*. By combining RNA-seq and DNase-seq they provide a comprehensive dataset of gene expression and chromatin accessibility changes during neuronal differentiation of deep-layer neurons and reveal that key genes linked to mature neuronal functions and bivalent genes in neural precursor cells become accessible during early differentiation. These findings underscore the crucial role of chromatin regulation in preparing neurons for maturation and unravel novel key insights into the regulatory mechanisms governing deep-layer neuronal differentiation.

Overall, this manuscript presents a novel technique for tracking neuron development from NPCs with specific birthdates. However, in its current form, it is largely descriptive and relies on correlative observations rather than elucidating a clear mechanism underlying chromatin and transcriptional changes. The provided data could be further leveraged to gain deeper insights into the molecular mechanisms governing deep-layer neuron development.

We would like to thank the reviewer for recognizing the methods used in this paper as 'a novel technique for tracking neuron development from NPCs with specific birthdates'. As the reviewer commented, this paper was descriptive, and we plan to prepare a revised version that includes results that approach 'the molecular mechanisms governing deep-layer neuron development' by analyzing the role of *Dmrt3* in neuronal differentiation, as shown in the response below, especially for point 9.

Major comments:

The authors have generated extensive RNA- and DNase-seq datasets across different developmental time points following birthdate labelling. However, the bioinformatics analyses and interpretations are limited and need further clarification and refinement:

1. The violin plots used to demonstrate expression and accessibility changes across developmental time points and the conclusions drawn from them are not convincing. The authors used a rank test to assess significant changes in expression, which only indicates the enrichment of genes with increased or decreased expression in each group. This cannot be directly interpreted as "significant upregulation." For instance, in Figures 4a and 4b, similar violin plots yield different statistical outcomes. The mean values on both graphs are comparable, yet Figure 4a suggests significant changes, while Figure 4b does not conclude significant downregulation of closing DHS genes. This is unconvincing. A more robust approach would be identifying DEGs between time points and analysing functional terms associated with these genes. The current plots do not support interpretations of gene upregulation, as each dot represents a gene, and the violin plot serves more as a population representation. The authors should either revisit their explanations and conclusions or include additional analyses and appropriate plots that support their claims of significant upregulation and downregulation of specific genes during development.

We would like to thank the reviewer for their helpful suggestions on presenting the data in Figure 4 more effectively. In future reanalysis, we will add an analysis focusing on DEGs, as suggested by the reviewer. Specifically, we will examine the overlap between DEGs identified by RNA-seq and genes with altered chromatin accessibility and test this using Fisher's exact test and other methods. This will allow us to verify the conclusions of this paper from multiple perspectives.

2. Figure 6b lacks clarity regarding the cutoff value used to categorise genes as K4me3 and K27me3 negative or positive from the heatmap. Even the "K4me3 negative" cluster displays a detectable signal of the mark, albeit at lower levels. Since only one plot of the entire gene body is provided, it is unclear what levels of enrichment are present, particularly at the promoter region. The authors are encouraged to provide additional informative plots and analyses of this ChIP-seq experiment, as this is a critical point where they draw conclusions about bivalent genes. This would not only strengthen their claims but could also uncover additional findings with more detailed analyses. A heatmap of clustered ChIP-seq signals of K4me3 and K27me3 alongside expression levels of the same genes (similar to Figure 2c) and differential accessibility (e.g., between NPC and E16) would better visualise and correlate histone modifications with chromatin

and gene expression states.

We would also like to thank this reviewer for their useful suggestions regarding Figure 6. In the next submission, we will try different methods to quantify H3K4me3 and H3K27me3 signals. Specifically, we plan to try methods using peak calling and methods that quantify signals in promoter regions.

We also plan to show new figures for changes in gene expression and chromatin accessibility in gene sets categorized by H3K4me3 and H3K27me3 signals.

3. The DNase-seq dataset can be better utilised to investigate differentially accessible motifs through development. Is this something the authors already looked into? This could strengthen mechanism investigation together with the ChIP-atlas results in Fig.6a

In the revised version, we will perform motif analysis and ChIP-atlas analysis for all genomic region sets showing differential accessibility. We will then use the results obtained to discuss the mechanisms of chromatin accessibility changes during the neuronal differentiation process in more depth.

4. The two distinct modes of H3K4me3 enrichment observed are not addressed and should be explained. Which genes belong to these two clusters? Is there a difference in DHS and gene expression between them?

In relation to point 2 of this reviewer, we will also re-analyze the differences in H3K4me3 patterns and changes in gene expression and chromatin accessibility. We believe that we can answer this reviewer's questions through the analyses using peak calling and signal quantification, as described in point 2.

5. The same concern regarding the use of violin plots to correlate gene expression with bivalent genes through development (Figure 6c) as mentioned earlier. It would be better to use DEGs and intersect them. This is particularly important given the wide range of gene expression levels in the already poised state.

In relation to this reviewer's point 1, we will also perform a reanalysis focusing on DEGs in Figure 6.

6. The authors limited their analyses to promoter/gene body regions. A survey of the bivalent marks and accessibility at enhancer regions would be also beneficial for understanding the changes at the chromatin landscape through development.

The results of Figure 3 showed that chromatin accessibility in the promoter region changes significantly during neuronal differentiation, and this paper has focused on the promoter region. However, as this reviewer has commented, we have realized that analysis of enhancers is also useful. We plan to re-analyze the changes in chromatin accessibility in the enhancer region for the revised version.

9. The mechanisms driving the activation and expression of poised neuronal genes through the development of deep-layer neurons is not uncovered. The authors suggest certain histone modifiers and the DNA methyltransferase Dnmt3 as potential drivers of chromatin landscape and transcriptional regulation changes; however, this remains speculative, as there is no direct evidence or validation of these factors binding to the identified target regions or changes in DNA methylation states. The authors should provide validation of their candidate factors' presence at potential targets, as well as changes in DNA methylation if they want to conclude these as the mechanisms driving deep-layer neuron development.

We thank the reviewer for pointing out the critical issue of the mechanism for the activation of poised genes. We agree that investigating the mechanism in more depth would improve our

paper.

To this end, we will analyze the role of Dmrt3, not Dnmt3, in activating poised genes. Dmrt3 is a transcription factor mainly involved in transcriptional repression, and our RNA-seq results indicate that it is highly expressed in NPCs, and its expression decreases during neuronal differentiation. Therefore, Dmrt3 may suppress poised genes in NPCs. Indeed, our preliminary results using public data have shown that knocking out Dmrt3 increased the expression of poised genes.

In future analyses, we plan to analyze the role of Dmrt3 using RNA-seq data from Dmrt3 knockout NPCs and Dmrt3 ChIP-seq data from NPCs.

Minor comments:

2. The motif analysis can be included in the main figures.

We appreciate the reviewer's positive suggestions. Regarding point 9, we will move the results of the motif analysis to the main figure after reanalysis about Dmrt3.

Reviewer #2 (Significance (Required)):

By introducing a novel genetic labelling method that tracks neurons based on their birthdates, the study provides a precise way to examine differentiation *in vivo*, adding valuable insights beyond traditional *in vitro* approaches. The combination of RNA-seq and DNase-seq analyses reveals how chromatin accessibility changes, particularly in bivalent genes, play a crucial role in neuronal maturation. This work highlights the importance of chromatin dynamics in establishing neuronal identity. The techniques and findings provide a useful framework for future studies, offering a path for deeper exploration of chromatin regulation across different neuronal types, stages of development, or disease contexts, making it a valuable contribution to the field of developmental neurobiology.

While the manuscript suggests the involvement of chromatin regulators such as Trithorax and Polycomb proteins, as well as Dnmt3 and DNA methylation, it lacks direct mechanistic evidence, such as ChIP-seq, bisulfite-seq, or loss-of-function experiments, to substantiate these claims.

The bioinformatics analyses and interpretations are limited and require further clarification and refinement.

The proposed mechanisms are not fully explored, leaving the manuscript largely descriptive rather than providing a detailed mechanistic understanding.

We would like to thank the reviewer again for their various suggestions for improving our manuscript. By performing the experimental plan described above, we try to resolve the reviewer's concerns and improve this paper.

Reviewer #3 (Evidence, reproducibility and clarity (Required)):

In this manuscript the authors use *in utero* electroporation of tamoxifen inducible reporters to permanently mark cortical neurons with a common birthdate. They then FACS harvest these cells for bulk DNase seq and RNA seq to see changes in chromatin regulation and gene expression as these newborn immature cortical neurons become deep layer neurons. As has been shown in prior studies that have addressed other neuronal types or used different methods to isolate developmental cell stages in the CNS, the authors find correlated changes between the opening or closing of chromatin with changes in gene expression. They use this information to localize chromatin marks that are associated with the differential expression of genes and conclude that many of the differential genes are bivalent for active and repressive chromatin marks. Finally the authors cross this dataset with a microarray they did of BDNF-inducible genes in cortical culture and suggest enrichment of this program in the differentially regulated gene set from *in vivo*.

Reviewer #3 (Significance (Required)):

The idea that chromatin regulation coordinates developmental changes in gene expression in neurons has been addressed with several different strategies over the past decade including prior strategies that allow for isolation of neurons with common birth dates. Many current strategies (well cited by the authors) use single cell sequencing and computational algorithms to deconvolve differentiation state from complex mixtures. This study takes an alternative approach to experimentally label these developmental stages which is nice to see for the validation of ground truth. However the study does not go far beyond current knowledge to use this method to add new concepts to the field. The main point of innovation seems to be the observation that the newborn neurons are primed at the chromatin level to express deep layer markers at the time they are born during embryonic life. This is useful to see but not unexpected on the basis of large scale single cell datasets. They also show that bivalent promoters prime developmental stage specific gene expression (in addition to the well-established function of this form of regulation in fate determination), however this too has been shown already in other neuron types.

We are very pleased that the reviewer evaluated our method as 'nice to see for the validation of ground truth' and distinguished it from the current mainstream method to trace the differentiation process computationally using single-cell analysis that tracks. On the other hand, we also agree with the reviewer's assessment that our results do not exceed previous knowledge. Therefore, as mentioned in our response to Reviewer #2, we plan to analyze the role of Dmrt3 in gene expression and chromatin structure during the neuronal differentiation process. This will allow us to clarify the novel insight into the neuronal differentiation process.

In addition to these conceptual limitations, there are some poorly supported comments in the text. For example, the fact that their microarray shows some genes in a category called "apoptosis" that are BDNF-sensitive does not meaningful suggest that BDNF induces excitotoxicity in embryonic cortical culture. BDNF has been well established as a survival factor for many kinds of neurons and is a common additive to serum-free media supplements (like B27). The appearance of "apoptosis" terms in the upregulated genes on the microarray more likely suggests either that the microarray is a poor detector of differential gene expression or that the genes in question are inaccurately categorized as "apoptotic" (GO terms are not terribly specific indicators of gene function). If the authors really wanted to test if BDNF was inducing apoptosis their cultures they could test this. However to use only the GO term data in such a strong statement about the biology of their system caused me to question the rigor of either their data or their analysis.

We are grateful to the reviewers for their important comments. We also agree that BDNF is an important neurotrophic factor and do not believe that it induces cell death. Therefore, we checked the following 40 genes, which showed chromatin closing from E12 to E16, upregulation upon BDNF stimulation, and the GO term 'programmed cell death'.

Cdip1, Diablo, Pla2g6, Braf, Tnfrsf25, Pa2g4, Mcl1, Hpn, Cebpb, EphA2, Plk3, Herpud1, Crip1, Dusp1, Sphk1, Irf5, Bag3, Stil, Fosl1, Cadm1, Lhx3, Hip1r, Relt, Irs2, Bmp8a, Ptcra, Mef2d, Prkcz, Rnf41, Pcid2

As a result, we found that there were no genes involved in the main pathway of apoptosis. From this, we understand that the GO terms related to cell death are listed in Figure 5f because 'the genes in question are inaccurately categorized as "apoptotic" ', as this reviewer pointed out.

We apologize for the misleading discussion in the previous manuscript and would like to thank the reviewer again for realizing this important point. We have corrected this in the new manuscript (page 9, line 263).

In addition, we will perform a reanalysis to confirm this conclusion of chromatin opening at neuronal activity-associated gene loci using public gene expression analysis data of neuronal stimulation.

A second example is the section about promoters being the focus of their discussion for DHS sites. Sure figure 3c shows promoters are more likely to be open compared with their contribution to the genome overall, but this is entirely expected since they are major gene TF binding sites, which is what DNase detects. However promoters do not look to be more likely to be differentially regulated over time (3c vs 3e), and the statement that promoters are more enriched in opening

compared with closing sites would require a statistical statement. Distal DHS sites appear equally more abundant in opening sites too.

We thank the reviewer for their thoughtful comments on our results. As the reviewer points out, the proportion of promoter regions in the opening DHS in Figure 3e is not so high compared to that in Figure 3c. However, as described in the Abstract and Introduction sections, we are interested in how neurons acquire their function during the differentiation process, and our main focus was on comparing neuron-specific and NPC-specific DHS here. In the comparison within Figure 3e, it is clear that the opening DHS has a higher proportion of promoter regions than the closing DHS. We made the necessary revisions to avoid any misunderstanding on this point (page 7, line 192).

On the other hand, as noted in the discussion, we are also interested in the role of the alteration in distal DHS. As in our response to Reviewer #2, we also plan to analyze changes in DHS in enhancer regions.

3. Description of the revisions that have already been incorporated in the transferred manuscript

Reviewer #1 (Evidence, reproducibility and clarity (Required)):

In Figure 1c, the actual values of the differentially expressed genes are unclear. Is this a Z-score? Please provide the log₂ expression values and specify the scale used for the heatmap and clustering.

We apologize for the unclear expression value of Figure 2c. As this reviewer pointed out, the heatmap shows the Z-score, and we provided the actual scale in the new figure.

Figure 5: It is somewhat unusual that the authors used microarray instead of RNA-seq for the BDNF stimulation of in vitro cortical neurons. Please provide a justification for this choice.

Gene expression analysis using microarrays is a well-established technique, though it is currently unfamiliar. Compared to RNA-seq, microarrays have the disadvantage that they can analyze only RNAs with probes and have a lower dynamic range. However, on the other hand, they have the advantages of reasonable cost and a simpler analysis method. In this paper, we performed microarray analysis for BDNF experiment, considering these advantages.

Figure 6: again, the data analyses are not comprehensively presented. What are the gene expression profiles of the other clusters (H3K27me₃⁺, H3K4me₃⁻/H3K27me₃⁻, H3K4me₃⁺)? Additionally, the sequencing data is inaccessible, and it is unclear how many samples (e.g., replicates) were used in this study for RNA-seq, DNase-seq, and ChIP-seq.

We apologize for the lack of gene expression patterns of other clusters in Figure 6c. We provided them in the new figure and confirmed that only bivalent genes (H3K4me₃⁺, H3K27me₃⁺) showed increased gene expression levels during neuronal differentiation and other clusters slight reduction (new Figure 6c). This result again suggests that the bivalent state in NPCs contributes to their activation during neuronal differentiation.

We described these data in the revised manuscript (page 10, line 296).

Raw sequence datasets (fastq files) and processed data were deposited in the DNA Data Bank of Japan (DDBJ) Sequence Read Archive, a partner of International Nucleotide Sequence Database Collaboration (INSDC), as already described in the Data Availability section. Although DDBJ does not provide a reviewer access system for raw sequence datasets, the reviewer's access to the processed data is as follows.

To review GEA accession E-GEAD-803, E-GEAD-859, E-GEAD-860:

- Go to <https://ddbj.nig.ac.jp/gea/reviewer>

- Enter accession E-GEAD-803, E-GEAD-859, or E-GEAD-860 and 20 characters access token into the

boxes

Please see the instructions below.

<https://www.ddbj.nig.ac.jp/gea/reviewer-access-e.html>

We will provide the access tokens in the final revised manuscript.

For replicate numbers, we apologize for forgetting to describe them for the BDNF microarray experiment, though those for RNA-seq, DNase-seq, and CHIP-seq were already described in the Methods section. The replicates numbers are as follows:

RNA-seq: two replicates

DNase-seq: two replicates

Microarray: three replicates

CHIP-seq: two replicates

We provided the replicate number of the microarray experiment in the revised manuscript (page 17, line 543).

Reviewer #2 (Evidence, reproducibility and clarity (Required)):

Major comments:

7. The authors begin by examining TFs enriched at E16 DHS regions and suggest that TrxG and PcG factors are highly enriched in neurons, initiating their investigation of bivalent marks. However, they later conclude that bivalent marks are present in the NPC state and later become accessible. It is unclear why PRC factors would be enriched at the neuronal stage when the authors conclude that the chromatin becomes more open (potentially by removal of K27me3). The authors should refine this section of the manuscript to better rationalise their methodology and results.

We are grateful to the reviewers for pointing out our poor explanation in Figure 6.

This section aimed to investigate the mechanism by which open genomic regions in E16 were established. We used CHIP-atlas to investigate the transcription factors enriched in the E16 DHS and found many of the components of TrxG and PcG in the previous experiments using ES cells, which are the stem cells as NPCs. Therefore, we hypothesized that binding both TrxG and PcG, meaning a bivalent state, in NPCs may be important for chromatin opening until E16. Therefore, we analyzed bivalent genes in NPCs rather than E16 neurons in Figure 6b-d.

We explained the rationale in detail in the revised version (page 9-10, line 269-288).

8. Do the authors find any expressional changes of the suggested candidate proteins at the RNA or protein levels through development?

We thank this reviewer for the useful suggestions. We agree that changes in the expression of TrxG and PcG components during neuronal differentiation are important information for considering the mechanism of chromatin structural changes in bivalent genes. Therefore, we checked the expression levels of genes encoding components of PcG or TrxG, determined by Schuettengruber et al., *Cell*, 2017, in our RNA-seq dataset (new Supplementary Data 5). More than half of them showed significant alteration, suggesting the possible contribution of alteration in the activity of PcG or TrxG or both on chromatin opening.

We described this point in the revised manuscript (page 12, line 370).

Minor comments:

1. The manuscript would improve with proofreading by a native English speaker.

We have already had proofreading by a native English speaker performed. We will also do it when submitting the revised version.

4. Description of analyses that authors prefer not to carry out

Reviewer #2 (Evidence, reproducibility and clarity (Required)):

One additional point, which may be beyond the scope of this paper, is that to demonstrate the temporal resolution of this birthdate tracking method robustly, the authors should also apply the technique to upper-layer neuron development and compare developmental differences that were previously challenging to capture due to lower resolution.

Reviewer #2 (Significance (Required)):

The study focuses exclusively on deep-layer excitatory neurons, without comparisons to other neuronal subtypes or non-neuronal cells. Including such comparisons would help determine whether the observed chromatin changes are unique to this specific population or part of a broader developmental process.

We are grateful for the reviewer's meaningful suggestions. We also think that by comparing with upper-layer neurons and non-neuronal cells, we can more comprehensively understand the development of the cerebral cortex. However, this paper primarily focuses on deep-layer neurons, and analysis of upper-layer neurons and non-neuronal cells will be future work.

We described this point in the revised manuscript (page 13, line 384).

First decision letter

MS ID#: dev.204564

MS Title: In vivo transition in chromatin accessibility during differentiation of deep-layer excitatory neurons in the neocortex

Authors: Seishin Sakai; Yurie Maeda; Keita Kawaji; Yutaka Suzuki; Yukiko Gotoh; Yusuke Kishi

Article Type: Review Commons Transfer

Dear Dr Kishi,

Thank you for sending your manuscript to Development through Review Commons.

As mentioned in my previous email, we would be happy to receive a revised of your manuscript along the lines suggested. Your revised paper will be re-reviewed by one or more of the original referees, and acceptance of your manuscript will depend on your addressing satisfactorily the reviewers' major concerns. Please also note that Development will normally permit only one round of major revision. If it would be helpful, you are welcome to contact us to discuss your revision in greater detail. Please send us a point-by-point response indicating your plans for addressing the referees' comments, and we will look over this and provide further guidance.

Please attend to all of the reviewers' comments and ensure that you clearly highlight all changes made in the revised manuscript. Please avoid using 'Tracked changes' in Word files as these are lost in PDF conversion. I should be grateful if you would also provide a point-by-point response detailing how you have dealt with the points raised by the reviewers in the 'Response to Reviewers' box. If you do not agree with any of their criticisms or suggestions please explain clearly why this is so.

First revisionAuthor response to reviewers' comments

1. General Statements [optional]

We sincerely thank the reviewers for their thorough evaluation and valuable suggestions, which have significantly helped us to identify areas for improvement in our manuscript.

In particular, we fully understand the concerns raised by Reviewers #1 and #2 regarding the insufficiency of data analysis. To address these critical concerns, we plan to perform more comprehensive analyses of our datasets in the revised version of the manuscript.

For the points raised by Reviewers #2 and #3 concerning the insufficient mechanistic insights and novelty of our study, we will conduct an in-depth investigation focusing on *Dmrt3*, aiming to elucidate its role in regulating chromatin accessibility and gene expression during neuronal differentiation. We have already initiated an analysis of RNA-seq data from *Dmrt3*-knockout embryonic brains, and preliminary results indicate that *Dmrt3* contributes to the regulation of bivalent genes in neural precursor cells. These findings provide compelling evidence for the involvement of *Dmrt3* in transcriptomic and chromatin regulation during neuronal differentiation, addressing the specific concerns raised by Reviewer #3.

Additionally, we will carefully revise the manuscript to address all other points raised by the reviewers, including minor suggestions.

Once again, we appreciate the reviewers' constructive feedback, and we are confident that these revisions will significantly strengthen the manuscript.

2. Description of the planned revisions

Reviewer #1 (Evidence, reproducibility and clarity (Required)):

The authors have developed a method for labeling a specific stage of differentiating neurons. Using this approach, they tracked the four-day differentiation process of deep-layer excitatory neurons in the mouse embryonic cortex. They investigated genome-wide changes in transcription patterns and chromatin accessibility using RNA-seq and DNase-seq. Additionally, they provided H3K4me3 and H3K27me3 ChIP-seq data from E12.0 NPCs. This resulting omics data would be a valuable resource for the field. While initial data analyses show potentially interesting findings, only part of the analyses are presented in the figures, lacking sufficient detail. Before publishing the manuscript, the authors should include more comprehensive analyses of their datasets. Specific suggestions are below.

We appreciate this reviewer's positive comments describing our study as 'a valuable resource for the field.' We plan to revise the paper, as noted below, to address this reviewer's concerns.

Figure 4 focuses on promoter-specific chromatin accessibility analysis. The author can process the data similarly to the transcription data. They should identify differentially accessible promoter regions across E13.0 to E16.0 and generate a heatmap with clustering. Additionally, the author should provide matched gene expression data, either in the form of a heatmap or box plot, corresponding to those differentially accessible promoter regions. Currently, Figure 4 only presents E16.0 data compared to E12.0, which is not comprehensive.

We thank the reviewer for the useful suggestions. In the following submission, we will determine gene sets for all chromatin accessibility change patterns, not just open/closed gene sets from E12 to E16. We will then illustrate the changes in gene expression for each gene set.

Reviewer #1 (Significance (Required)):

Multi-omics data from the differentiation process of deep-layer excitatory neurons would be a valuable resource for the field.

Once again, we would like to thank the reviewers for their positive comments.

Reviewer #2 (Evidence, reproducibility and clarity (Required)):

Summary:

The manuscript from Sakai et al. examines changes in chromatin accessibility during the differentiation of deep-layer excitatory neurons in the neocortex. The authors establish a novel genetic labelling method that tracks differentiating neurons based on their birthdates allowing following neuronal differentiation *in vivo*. By combining RNA-seq and DNase-seq they provide a comprehensive dataset of gene expression and chromatin accessibility changes during neuronal differentiation of deep-layer neurons and reveal that key genes linked to mature neuronal functions and bivalent genes in neural precursor cells become accessible during early differentiation. These findings underscore the crucial role of chromatin regulation in preparing neurons for maturation and unravel novel key insights into the regulatory mechanisms governing deep-layer neuronal differentiation.

Overall, this manuscript presents a novel technique for tracking neuron development from NPCs with specific birthdates. However, in its current form, it is largely descriptive and relies on correlative observations rather than elucidating a clear mechanism underlying chromatin and transcriptional changes. The provided data could be further leveraged to gain deeper insights into the molecular mechanisms governing deep-layer neuron development.

We would like to thank the reviewer for recognizing the methods used in this paper as 'a novel technique for tracking neuron development from NPCs with specific birthdates'. As the reviewer commented, this paper was descriptive, and we plan to prepare a revised version that includes results that approach 'the molecular mechanisms governing deep-layer neuron development' by analyzing the role of *Dmrt3* in neuronal differentiation, as shown in the response below, especially for point 9.

Major comments:

The authors have generated extensive RNA- and DNase-seq datasets across different developmental time points following birthdate labelling. However, the bioinformatics analyses and interpretations are limited and need further clarification and refinement:

1. The violin plots used to demonstrate expression and accessibility changes across developmental time points and the conclusions drawn from them are not convincing. The authors used a rank test to assess significant changes in expression, which only indicates the enrichment of genes with increased or decreased expression in each group. This cannot be directly interpreted as "significant upregulation." For instance, in Figures 4a and 4b, similar violin plots yield different statistical outcomes. The mean values on both graphs are comparable, yet Figure 4a suggests significant changes, while Figure 4b does not conclude significant downregulation of closing DHS genes. This is unconvincing. A more robust approach would be identifying DEGs between time points and analysing functional terms associated with these genes. The current plots do not support interpretations of gene upregulation, as each dot represents a gene, and the violin plot serves more as a population representation. The authors should either revisit their explanations and conclusions or include additional analyses and appropriate plots that support their claims of significant upregulation and downregulation of specific genes during development.

We would like to thank the reviewer for their helpful suggestions on presenting the data in Figure 4 more effectively. In future reanalysis, we will add an analysis focusing on DEGs, as suggested by the reviewer. Specifically, we will examine the overlap between DEGs identified by RNA-seq and genes with altered chromatin accessibility and test this using Fisher's exact test and other methods. This will allow us to verify the conclusions of this paper from multiple perspectives.

2. Figure 6b lacks clarity regarding the cutoff value used to categorise genes as K4me3 and K27me3 negative or positive from the heatmap. Even the "K4me3 negative" cluster displays a detectable signal of the mark, albeit at lower levels. Since only one plot of the entire gene body is provided, it is unclear what levels of enrichment are present, particularly at the promoter region. The authors are encouraged to provide additional informative plots and analyses of this ChIP-seq experiment, as this is a critical point where they draw conclusions about bivalent genes. This would not only strengthen their claims but could also uncover additional findings with more detailed analyses. A heatmap of clustered ChIP-seq signals of K4me3 and K27me3 alongside expression levels of the same genes (similar to Figure 2c) and differential accessibility (e.g., between NPC and E16) would better visualise and correlate histone modifications with chromatin and gene expression states.

We would also like to thank this reviewer for their useful suggestions regarding Figure 6. In the next submission, we will try different methods to quantify H3K4me3 and H3K27me3 signals. Specifically, we plan to try methods using peak calling and methods that quantify signals in promoter regions.

We also plan to show new figures for changes in gene expression and chromatin accessibility in gene sets categorized by H3K4me3 and H3K27me3 signals.

3. The DNase-seq dataset can be better utilised to investigate differentially accessible motifs through development. Is this something the authors already looked into? This could strengthen mechanism investigation together with the ChIP-atlas results in Fig.6a

In the revised version, we will perform motif analysis and ChIP-atlas analysis for all genomic region sets showing differential accessibility. We will then use the results obtained to discuss the mechanisms of chromatin accessibility changes during the neuronal differentiation process in more depth.

4. The two distinct modes of H3K4me3 enrichment observed are not addressed and should be explained. Which genes belong to these two clusters? Is there a difference in DHS and gene expression between them?

In relation to point 2 of this reviewer, we will also re-analyze the differences in H3K4me3 patterns and changes in gene expression and chromatin accessibility. We believe that we can answer this reviewer's questions through the analyses using peak calling and signal quantification, as described in point 2.

5. The same concern regarding the use of violin plots to correlate gene expression with bivalent genes through development (Figure 6c) as mentioned earlier. It would be better to use DEGs and intersect them. This is particularly important given the wide range of gene expression levels in the already poised state.

In relation to this reviewer's point 1, we will also perform a reanalysis focusing on DEGs in Figure 6.

6. The authors limited their analyses to promoter/gene body regions. A survey of the bivalent marks and accessibility at enhancer regions would be also beneficial for understanding the changes at the chromatin landscape through development.

The results of Figure 3 showed that chromatin accessibility in the promoter region changes significantly during neuronal differentiation, and this paper has focused on the promoter region. However, as this reviewer has commented, we have realized that analysis of enhancers is also useful. We plan to re-analyze the changes in chromatin accessibility in the enhancer region for the revised version.

9. The mechanisms driving the activation and expression of poised neuronal genes through the development of deep-layer neurons is not uncovered. The authors suggest certain histone modifiers and the DNA methyltransferase Dnmt3 as potential drivers of chromatin landscape and transcriptional regulation changes; however, this remains speculative, as there is no direct evidence or validation of these factors binding to the identified target regions or changes in DNA methylation states. The authors should provide validation of their candidate factors' presence at potential targets, as well as changes in DNA methylation if they want to conclude these as the mechanisms driving deep-layer neuron development.

We thank the reviewer for pointing out the critical issue of the mechanism for the activation of poised genes. We agree that investigating the mechanism in more depth would improve our paper.

To this end, we will analyze the role of Dmrt3, not Dnmt3, in activating poised genes. Dmrt3 is a transcription factor mainly involved in transcriptional repression, and our RNA-seq results indicate that it is highly expressed in NPCs, and its expression decreases during neuronal differentiation. Therefore, Dmrt3 may suppress poised genes in NPCs. Indeed, our preliminary results using public data have shown that knocking out Dmrt3 increased the expression of poised genes.

In future analyses, we plan to analyze the role of Dmrt3 using RNA-seq data from Dmrt3 knockout NPCs and Dmrt3 ChIP-seq data from NPCs.

Minor comments:

2. The motif analysis can be included in the main figures.

We appreciate the reviewer's positive suggestions. Regarding point 9, we will move the results of the motif analysis to the main figure after reanalysis about Dmrt3.

Reviewer #2 (Significance (Required)):

By introducing a novel genetic labelling method that tracks neurons based on their birthdates, the study provides a precise way to examine differentiation in vivo, adding valuable insights beyond traditional in vitro approaches. The combination of RNA-seq and DNase-seq analyses reveals how chromatin accessibility changes, particularly in bivalent genes, play a crucial role in neuronal maturation. This work highlights the importance of chromatin dynamics in establishing neuronal identity. The techniques and findings provide a useful framework for future studies, offering a path for deeper exploration of chromatin regulation across different neuronal types, stages of development, or disease contexts, making it a valuable contribution to the field of developmental neurobiology.

While the manuscript suggests the involvement of chromatin regulators such as Trithorax and Polycomb proteins, as well as Dnmt3 and DNA methylation, it lacks direct mechanistic evidence, such as ChIP-seq, bisulfite-seq, or loss-of-function experiments, to substantiate these claims.

The bioinformatics analyses and interpretations are limited and require further clarification and refinement.

The proposed mechanisms are not fully explored, leaving the manuscript largely descriptive rather than providing a detailed mechanistic understanding.

We would like to thank the reviewer again for their various suggestions for improving our manuscript. By performing the experimental plan described above, we try to resolve the reviewer's concerns and improve this paper.

Reviewer #3 (Evidence, reproducibility and clarity (Required)):

In this manuscript the authors use in utero electroporation of tamoxifen inducible reporters to permanently mark cortical neurons with a common birthdate. They then FACS harvest these cells for bulk DNase seq and RNA seq to see changes in chromatin regulation and gene expression as these newborn immature cortical neurons become deep layer neurons. As has been shown in prior studies that have addressed other neuronal types or used different methods to isolate developmental cell stages in the CNS, the authors find correlated changes between the opening or closing of chromatin with changes in gene expression. They use this information to localize chromatin marks that are associated with the differential expression of genes and conclude that many of the differential genes are bivalent for active and repressive chromatin marks. Finally the authors cross this dataset with a microarray they did of BDNF-inducible genes in cortical culture and suggest enrichment of this program in the differentially regulated gene set from in vivo.

Reviewer #3 (Significance (Required)):

The idea that chromatin regulation coordinates developmental changes in gene expression in neurons has been addressed with several different strategies over the past decade including prior strategies that allow for isolation of neurons with common birth dates. Many current strategies (well cited by the authors) use single cell sequencing and computational algorithms to deconvolve differentiation state from complex mixtures. This study takes an alternative approach to experimentally label these developmental stages which is nice to see for the validation of ground truth. However the study does not go far beyond current knowledge to use this method to add new

concepts to the field. The main point of innovation seems to be the observation that the newborn neurons are primed at the chromatin level to express deep layer markers at the time they are born during embryonic life. This is useful to see but not unexpected on the basis of large scale single cell datasets. They also show that bivalent promoters prime developmental stage specific gene expression (in addition to the well-established function of this form of regulation in fate determination), however this too has been shown already in other neuron types.

We are very pleased that the reviewer evaluated our method as 'nice to see for the validation of ground truth' and distinguished it from the current mainstream method to trace the differentiation process computationally using single-cell analysis that tracks. On the other hand, we also agree with the reviewer's assessment that our results do not exceed previous knowledge. Therefore, as mentioned in our response to Reviewer #2, we plan to analyze the role of Dmrt3 in gene expression and chromatin structure during the neuronal differentiation process. This will allow us to clarify the novel insight into the neuronal differentiation process.

In addition to these conceptual limitations, there are some poorly supported comments in the text. For example, the fact that their microarray shows some genes in a category called "apoptosis" that are BDNF-sensitive does not meaningful suggest that BDNF induces excitotoxicity in embryonic cortical culture. BDNF has been well established as a survival factor for many kinds of neurons and is a common additive to serum-free media supplements (like B27). The appearance of "apoptosis" terms in the upregulated genes on the microarray more likely suggests either that the microarray is a poor detector of differential gene expression or that the genes in question are inaccurately categorized as "apoptotic" (GO terms are not terribly specific indicators of gene function). If the authors really wanted to test if BDNF was inducing apoptosis their cultures they could test this. However to use only the GO term data in such a strong statement about the biology of their system caused me to question the rigor of either their data or their analysis.

We are grateful to the reviewers for their important comments. We also agree that BDNF is an important neurotrophic factor and do not believe that it induces cell death. Therefore, we checked the following 40 genes, which showed chromatin closing from E12 to E16, upregulation upon BDNF stimulation, and the GO term 'programmed cell death'.

Cdip1, Diablo, Pla2g6, Braf, Tnfrsf25, Pa2g4, Mcl1, Hpn, Cebpb, Epha2, Plk3, Herpud1, Crip1, Dusp1, Sphk1, Irf5, Bag3, Stil, Fosl1, Cadm1, Lhx3, Hip1r, Relt, Irs2, Bmp8a, Ptcra, Mef2d, Prkcz, Rnf41, Pcid2

As a result, we found that there were no genes involved in the main pathway of apoptosis. From this, we understand that the GO terms related to cell death are listed in Figure 5f because 'the genes in question are inaccurately categorized as "apoptotic" ', as this reviewer pointed out.

We apologize for the misleading discussion in the previous manuscript and would like to thank the reviewer again for realizing this important point. We have corrected this in the new manuscript (page 9, line 263).

In addition, we will perform a reanalysis to confirm this conclusion of chromatin opening at neuronal activity-associated gene loci using public gene expression analysis data of neuronal stimulation.

A second example is the section about promoters being the focus of their discussion for DHS sites. Sure figure 3c shows promoters are more likely to be open compared with their contribution to the genome overall, but this is entirely expected since they are major gene TF binding sites, which is what DNase detects. However promoters do not look to be more likely to be differentially regulated over time (3c vs 3e), and the statement that promoters are more enriched in opening compared with closing sites would require a statistical statement. Distal DHS sites appear equally more abundant in opening sites too.

We thank the reviewer for their thoughtful comments on our results. As the reviewer points out, the proportion of promoter regions in the opening DHS in Figure 3e is not so high compared to that in Figure 3c. However, as described in the Abstract and Introduction sections, we are interested in how neurons acquire their function during the differentiation process, and our main focus was on comparing neuron-specific and NPC-specific DHS here. In the comparison within Figure 3e, it is clear that the opening DHS has a higher proportion of promoter regions than the closing DHS. We made the necessary revisions to avoid any misunderstanding on this point (page 7, line 192).

On the other hand, as noted in the discussion, we are also interested in the role of the alteration in distal DHS. As in our response to Reviewer #2, we also plan to analyze changes in DHS in enhancer regions.

3. Description of the revisions that have already been incorporated in the transferred manuscript

Reviewer #1 (Evidence, reproducibility and clarity (Required)):

In Figure 1c, the actual values of the differentially expressed genes are unclear. Is this a Z-score? Please provide the log₂ expression values and specify the scale used for the heatmap and clustering.

We apologize for the unclear expression value of Figure 2c. As this reviewer pointed out, the heatmap shows the Z-score, and we provided the actual scale in the new figure.

Figure 5: It is somewhat unusual that the authors used microarray instead of RNA-seq for the BDNF stimulation of in vitro cortical neurons. Please provide a justification for this choice.

Gene expression analysis using microarrays is a well-established technique, though it is currently unfamiliar. Compared to RNA-seq, microarrays have the disadvantage that they can analyze only RNAs with probes and have a lower dynamic range. However, on the other hand, they have the advantages of reasonable cost and a simpler analysis method. In this paper, we performed microarray analysis for BDNF experiment, considering these advantages.

Figure 6: again, the data analyses are not comprehensively presented. What are the gene expression profiles of the other clusters (H3K27me₃⁺, H3K4me₃⁻/H3K27me₃⁻, H3K4me₃⁺)? Additionally, the sequencing data is inaccessible, and it is unclear how many samples (e.g., replicates) were used in this study for RNA-seq, DNase-seq, and ChIP-seq.

We apologize for the lack of gene expression patterns of other clusters in Figure 6c. We provided them in the new figure and confirmed that only bivalent genes (H3K4me₃⁺, H3K27me₃⁺) showed increased gene expression levels during neuronal differentiation and other clusters slight reduction (new Figure 6c). This result again suggests that the bivalent state in NPCs contributes to their activation during neuronal differentiation.

We described these data in the revised manuscript (page 10, line 296).

Raw sequence datasets (fastq files) and processed data were deposited in the DNA Data Bank of Japan (DDBJ) Sequence Read Archive, a partner of International Nucleotide Sequence Database Collaboration (INSDC), as already described in the Data Availability section. Although DDBJ does not provide a reviewer access system for raw sequence datasets, the reviewer's access to the processed data is as follows.

To review GEA accession E-GEAD-803, E-GEAD-859, E-GEAD-860:

- Go to <https://ddbj.nig.ac.jp/gea/reviewer>

- Enter accession E-GEAD-803, E-GEAD-859, or E-GEAD-860 and 20 characters access token into the boxes

Please see the instructions below.

<https://www.ddbj.nig.ac.jp/gea/reviewer-access-e.html>

We will provide the access tokens in the final revised manuscript.

For replicate numbers, we apologize for forgetting to describe them for the BDNF microarray experiment, though those for RNA-seq, DNase-seq, and ChIP-seq were already described in the Methods section. The replicates numbers are as follows:

RNA-seq: two replicates

DNase-seq: two replicates

Microarray: three replicates

ChIP-seq: two replicates

We provided the replicate number of the microarray experiment in the revised manuscript (page 17, line 543).

Reviewer #2 (Evidence, reproducibility and clarity (Required)):

Major comments:

7. The authors begin by examining TFs enriched at E16 DHS regions and suggest that TrxG and PcG factors are highly enriched in neurons, initiating their investigation of bivalent marks. However, they later conclude that bivalent marks are present in the NPC state and later become accessible. It is unclear why PRC factors would be enriched at the neuronal stage when the authors conclude that the chromatin becomes more open (potentially by removal of K27me3). The authors should refine this section of the manuscript to better rationalise their methodology and results.

We are grateful to the reviewers for pointing out our poor explanation in Figure 6.

This section aimed to investigate the mechanism by which open genomic regions in E16 were established. We used ChIP-atlas to investigate the transcription factors enriched in the E16 DHS and found many of the components of TrxG and PcG in the previous experiments using ES cells, which are the stem cells as NPCs. Therefore, we hypothesized that binding both TrxG and PcG, meaning a bivalent state, in NPCs may be important for chromatin opening until E16. Therefore, we analyzed bivalent genes in NPCs rather than E16 neurons in Figure 6b-d.

We explained the rationale in detail in the revised version (page 9-10, line 269-288).

8. Do the authors find any expressional changes of the suggested candidate proteins at the RNA or protein levels through development?

We thank this reviewer for the useful suggestions. We agree that changes in the expression of TrxG and PcG components during neuronal differentiation are important information for considering the mechanism of chromatin structural changes in bivalent genes. Therefore, we checked the expression levels of genes encoding components of PcG or TrxG, determined by Schuettengruber et al., *Cell*, 2017, in our RNA-seq dataset (new Supplementary Data 5). More than half of them showed significant alteration, suggesting the possible contribution of alteration in the activity of PcG or TrxG or both on chromatin opening.

We described this point in the revised manuscript (page 12, line 370).

Minor comments:

1. The manuscript would improve with proofreading by a native English speaker.

We have already had proofreading by a native English speaker performed. We will also do it when submitting the revised version.

4. Description of analyses that authors prefer not to carry out

Reviewer #2 (Evidence, reproducibility and clarity (Required)):

One additional point, which may be beyond the scope of this paper, is that to demonstrate the temporal resolution of this birthdate tracking method robustly, the authors should also apply the technique to upper-layer neuron development and compare developmental differences that were previously challenging to capture due to lower resolution.

Reviewer #2 (Significance (Required)):

The study focuses exclusively on deep-layer excitatory neurons, without comparisons to other neuronal subtypes or non-neuronal cells. Including such comparisons would help determine whether the observed chromatin changes are unique to this specific population or part of a broader developmental process.

We are grateful for the reviewer's meaningful suggestions. We also think that by comparing with upper-layer neurons and non-neuronal cells, we can more comprehensively understand the development of the cerebral cortex. However, this paper primarily focuses on deep-layer neurons, and analysis of upper-layer neurons and non-neuronal cells will be future work.

We described this point in the revised manuscript (page 13, line 384).

Second decision letter

MS ID#: dev.204564R1

MS Title: In vivo transition in chromatin accessibility during differentiation of deep-layer excitatory neurons in the neocortex

Authors: Seishin Sakai; Yurie Maeda; Keita Kawaji; Yutaka Suzuki; Yukiko Gotoh; Yusuke Kishi

Article Type: Review Commons Transfer

Dear Dr Kishi,

Many thanks for transferring your manuscript to Development from Review Commons. I have now had the chance to review your documents, and we would like to invite you to revise your manuscript according to your revision plan. I encourage you to include all the data analysis you outline in your revision plan in your revised manuscript. Once we receive the revised version, I will seek input from the Review Commons referees. If you have any questions about this process, please do get in touch.

Second revision

Author response to reviewers' comments

We sincerely appreciate the reviewers' thorough evaluation and valuable suggestions, which have greatly helped us refine our manuscript.

In particular, we fully acknowledge the concerns raised by Reviewers #1 and #2 regarding the insufficiency of data analysis. To address these critical issues, we have performed a more comprehensive analysis of our datasets in the revised manuscript, incorporating new Figures 4, 6, and 7, Supplementary Figures 1, and 2, as well as Supplementary Data 1-8.

Regarding the concerns raised by Reviewers #2 and #3 about the mechanistic insights and novelty of our study, we have conducted an in-depth investigation focusing on Dmrt3 and Dmrt2 to elucidate their association with chromatin accessibility and gene expression during neuronal differentiation (new Figure 7). These findings indicate the involvement of Dmrt3 in transcriptomic and chromatin regulation during neuronal differentiation, directly addressing the specific concerns raised by Reviewer #3.

Additionally, we have carefully revised the manuscript to incorporate all other comments from the reviewers, including minor suggestions.

Once again, we deeply appreciate the reviewers' constructive feedback, and we are confident that these revisions have significantly strengthened our manuscript.

Reviewer #1 (Evidence, reproducibility and clarity (Required)):

The authors have developed a method for labeling a specific stage of differentiating neurons. Using this approach, they tracked the four-day differentiation process of deep-layer excitatory neurons in the mouse embryonic cortex. They investigated genome-wide changes in transcription patterns and chromatin accessibility using RNA-seq and DNase-seq. Additionally, they provided H3K4me3 and H3K27me3 CHIP-seq data from E12.0 NPCs. This resulting omics data would be a valuable resource

for the field. While initial data analyses show potentially interesting findings, only part of the analyses are presented in the figures, lacking sufficient detail. Before publishing the manuscript, the authors should include more comprehensive analyses of their datasets. Specific suggestions are below.

We sincerely appreciate the reviewer's positive feedback, describing our study as "a valuable resource for the field." In response to the reviewer's concerns, we have carefully revised our manuscript accordingly.

In Figure 1c, the actual values of the differentially expressed genes are unclear. Is this a Z-score? Please provide the log₂ expression values and specify the scale used for the heatmap and clustering.

We apologize for the lack of clarity in the expression values presented in Figure 2c. As the reviewer correctly pointed out, the heatmap displays Z-scores. In the revised manuscript, we have included the actual scale in the updated figure (new Figure 2c).

Figure 4 focuses on promoter-specific chromatin accessibility analysis. The author can process the data similarly to the transcription data. They should identify differentially accessible promoter regions across E13.0 to E16.0 and generate a heatmap with clustering. Additionally, the author should provide matched gene expression data, either in the form of a heatmap or box plot, corresponding to those differentially accessible promoter regions. Currently, Figure 4 only presents E16.0 data compared to E12.0, which is not comprehensive.

We sincerely thank the reviewer for their valuable suggestions. Considering the response to Point 1 raised by Reviewer #2, we analyzed the association between differentially expressed genes (DEGs) and DHSs across E12.0 to E16.0, rather than limiting the comparison to E12.0 and E16.0. To assess the significance of these overlaps, we performed Fisher's exact test (new Figure 4A).

This analysis not only provides a comprehensive view of the relationship between gene expression changes and chromatin accessibility but also highlights that among promoter-opening genes from E12.0 to E16.0, those that opened exclusively at E16.0 were strongly associated with gene activation.

Furthermore, we conducted a similar analysis to examine the associations between DEGs, DHSs, layer-specific gene expression, bivalent genes, and Dmrt-regulating genes, offering a more comprehensive insight into our dataset (new Figures 6 and 7, Supplementary Figures 1 and 2).

We described these data in the revised manuscript (page 7, line 200; page 9, line 267; page 10, line 325; page 11, line 361).

Figure 5: It is somewhat unusual that the authors used microarray instead of RNA-seq for the BDNF stimulation of in vitro cortical neurons. Please provide a justification for this choice.

Gene expression analysis using microarrays is a well-established technique, though it has become less commonly used in recent years. Compared to RNA-seq, microarrays have the disadvantage of being limited to analyzing only RNAs with available probes and having a lower dynamic range. However, they offer advantages such as cost-effectiveness and a simpler analytical process. In this study, we chose to perform microarray analysis for the BDNF experiment, taking these advantages into consideration.

Figure 6: again, the data analyses are not comprehensively presented. What are the gene expression profiles of the other clusters (H3K27me₃⁺, H3K4me₃⁻/H3K27me₃⁻, H3K4me₃⁺)?

We apologize for the omission of gene expression patterns for other clusters in Figure 6C. In the revised manuscript, we have included these patterns in the updated figure (new Figure 6C). Our analysis confirmed that only bivalent genes (H3K4me₃⁺ H3K27me₃⁺) exhibited increased expression levels during neuronal differentiation, while other clusters showed a slight reduction. This finding further supports the idea that the bivalent state in NPCs contributes to their activation during neuronal differentiation.

We described these data in the revised manuscript (page 10, line 329).

Additionally, the sequencing data is inaccessible, and it is unclear how many samples (e.g., replicates) were used in this study for RNA-seq, DNase-seq, and ChIP-seq.

Raw sequence datasets (FASTQ files) and processed data have been deposited in the DNA Data Bank of Japan (DDBJ) Sequence Read Archive, a partner of the International Nucleotide Sequence Database Collaboration (INSDC), as stated in the Data Availability section. While DDBJ does not offer a reviewer access system for raw sequence datasets, the reviewer can access the processed data as follows:

To review GEA accession E-GEAD-803, E-GEAD-859, E-GEAD-860:

- Go to <https://ddbj.nig.ac.jp/gea/reviewer>

- Enter accession E-GEAD-803, E-GEAD-859, or E-GEAD-860 and 20 characters access token into the boxes

Please see the instructions below.

<https://www.ddbj.nig.ac.jp/gea/reviewer-access-e.html>

Access Tokens:

- E-GEAD-803: 6IBwEXUbEUYoK4z2EjsA
- E-GEAD-859: KLEnsOt9oRrgjdsEsj1f
- E-GEAD-860: yQ3qBs4GVW07Gaj1n2sk

*The expired date of the tokens: 11th July 2025.

Regarding the number of replicates, we apologize for the omission of this information for the BDNF microarray experiment. The replicate numbers for RNA-seq, DNase-seq, and ChIP-seq were already provided in the previous Methods section. The details are as follows:

- RNA-seq: Two replicates
- DNase-seq: Two replicates
- Microarray: Three replicates
- ChIP-seq: Two replicates

We have now included the replicate number for the microarray experiment in the revised manuscript (page 33, line 1091, 1096; page 34, line 1125, 1136).

Reviewer #1 (Significance (Required)):

Multi-omics data from the differentiation process of deep-layer excitatory neurons would be a valuable resource for the field.

Once again, we would like to thank this reviewer for the positive comments.

Reviewer #2 (Evidence, reproducibility and clarity (Required)):

Summary:

The manuscript from Sakai et al. examines changes in chromatin accessibility during the differentiation of deep-layer excitatory neurons in the neocortex. The authors establish a novel genetic labelling method that tracks differentiating neurons based on their birthdates allowing following neuronal differentiation in vivo. By combining RNA-seq and DNase-seq they provide a comprehensive dataset of gene expression and chromatin accessibility changes during neuronal differentiation of deep-layer neurons and reveal that key genes linked to mature neuronal functions and bivalent genes in neural precursor cells become accessible during early differentiation. These findings underscore the crucial role of chromatin regulation in preparing neurons for maturation and unravel novel key insights into the regulatory mechanisms governing deep-layer neuronal differentiation.

Overall, this manuscript presents a novel technique for tracking neuron development from NPCs with specific birthdates. However, in its current form, it is largely descriptive and relies on correlative observations rather than elucidating a clear mechanism underlying chromatin and transcriptional changes. The provided data could be further leveraged to gain deeper insights into the molecular mechanisms governing deep-layer neuron development.

We sincerely appreciate the reviewer's recognition of our methods as "a novel technique for tracking neuron development from NPCs with specific birthdates." As noted, the previous version was primarily descriptive. To address this, we have incorporated results that explore "the molecular mechanisms governing deep-layer neuron development" by analyzing the roles of *Dmrt3* and *Dmrta2* in neuronal differentiation, particularly in response to point 9.

One additional point, which may be beyond the scope of this paper, is that to demonstrate the temporal resolution of this birthdate tracking method robustly, the authors should also apply the technique to upper-layer neuron development and compare developmental differences that were previously challenging to capture due to lower resolution.

We sincerely appreciate the reviewer's insightful suggestions. We agree that comparing deep-layer neurons with upper-layer neurons and non-neuronal cells would provide a more comprehensive understanding of cerebral cortex development. However, as this study primarily focuses on deep-layer neurons, the analysis of upper-layer neurons and non-neuronal cells will be addressed in future work.

We described this point in the revised manuscript as a limitation of this study (page 15, line 513).

Major comments:

The authors have generated extensive RNA- and DNase-seq datasets across different developmental time points following birthdate labelling. However, the bioinformatics analyses and interpretations are limited and need further clarification and refinement:

1. The violin plots used to demonstrate expression and accessibility changes across developmental time points and the conclusions drawn from them are not convincing. The authors used a rank test to assess significant changes in expression, which only indicates the enrichment of genes with increased or decreased expression in each group. This cannot be directly interpreted as "significant upregulation." For instance, in Figures 4a and 4b, similar violin plots yield different statistical outcomes. The mean values on both graphs are comparable, yet Figure 4a suggests significant changes, while Figure 4b does not conclude significant downregulation of closing DHS genes. This is unconvincing. A more robust approach would be identifying DEGs between time points and analysing functional terms associated with these genes. The current plots do not support interpretations of gene upregulation, as each dot represents a gene, and the violin plot serves more as a population representation. The authors should either revisit their explanations and conclusions or include additional analyses and appropriate plots that support their claims of significant upregulation and downregulation of specific genes during development.

We would like to thank the reviewer for their helpful suggestions on presenting the data in Figure 4 more effectively. Based on this feedback, we have analyzed the association between DEGs and DHSs from E12.0 to E16.0, rather than limiting the comparison between E12.0 and E16.0, and performed Fisher's exact test to determine the significance of the overlaps (new Figure 4A).

This analysis supports the observation that promoter opening during neuronal differentiation—especially opening at E16.0—contributes to gene activation, while promoter closing does not significantly contribute to gene repression. Additionally, we performed GO analysis to identify the functional terms associated with genes that have DHSs at their promoter regions at each stage. This analysis revealed that genes with promoters opening exclusively at E16.0 were linked to neuronal functions, such as synapse and axon formation (new Figure 4C, Supplementary Data 2).

We again appreciate the reviewer's insightful comment, which helped improve the presentation and interpretation of our data.

We described this point in the revised manuscript (page 7, line 200).

2. Figure 6b lacks clarity regarding the cutoff value used to categorise genes as K4me3 and K27me3 negative or positive from the heatmap. Even the "K4me3 negative" cluster displays a detectable signal of the mark, albeit at lower levels. Since only one plot of the entire gene body is provided, it is unclear what levels of enrichment are present, particularly at the promoter region. The authors are encouraged to provide additional informative plots and analyses of this ChIP-seq experiment, as this is a critical point where they draw conclusions about bivalent genes. This would not only strengthen their claims but could also uncover additional findings with more detailed analyses. A heatmap of clustered ChIP-seq signals of K4me3 and K27me3 alongside expression levels of the same genes (similar to Figure 2c) and differential accessibility (e.g., between NPC and E16) would better visualise and correlate histone modifications with chromatin and gene expression states.

We would also like to thank the reviewer for their useful suggestions regarding Figure 6. In response to the comment, we performed peak calling with a clearer cutoff value to identify H3K4me3- and/or H3K27me3-positive genes at their promoter regions (new Figure 6B), instead of using the ngsplot software as in the previous manuscript. We have also included additional data showing the relationship between bivalent gene sets and chromatin accessibility (new Figure 6C), as well as gene expression levels (Figure 6D and Supplementary Figure 2).

We described this point in the revised manuscript (page 10, line 326).

3. The DNase-seq dataset can be better utilised to investigate differentially accessible motifs through development. Is this something the authors already looked into? This could strengthen mechanism investigation together with the ChIP-atlas results in Fig.6a

We are grateful for the reviewer's valuable advice. In the revised manuscript, we have included ChIP-atlas and HOMER motif analyses for all or specific DHSs at each stage (Supplementary Data 3-7). These analyses suggest the involvement of well-known transcription factors for neural development and neurodevelopmental diseases, which are listed in the Results section (page 8, line 235; page 9, line 273; page 10, line 298), in regulating chromatin accessibility. Additionally, the HOMER motif analysis for E16.0 DHSs highlighted Dmrt3 and Dmrt2 as potential candidate factors regulating chromatin opening during neuronal differentiation, as detailed in our response to point 9 of this reviewer.

We described this point in the revised manuscript (page 111, line 348).

4. The two distinct modes of H3K4me3 enrichment observed are not addressed and should be explained. Which genes belong to these two clusters? Is there a difference in DHS and gene expression between them?

Based on Point 2 of this reviewer, we reanalyzed the H3K4me3 and H3K27me3 ChIP-seq data using a peak calling method, and therefore, we have not included the distinction between the two modes of H3K4me3 in the revised manuscript.

The two distinct modes of H3K4me3 distribution in the previous figure correspond to TSS-enriched and broad H3K4me3 domains, as described by Benayoun et al., *Cell*, 2014. Generally, broad H3K4me3 domains are associated with cell identity genes and contribute to efficient transcription and stable expression with lower variability between individual cells. While it is possible that these two gene sets exhibit different gene expression levels and chromatin accessibility in our data, we did not analyze them further, as the main purpose of the ChIP-seq was to identify bivalent genes.

Nonetheless, we truly appreciate this reviewer's thoughtful comment, which has helped refine our interpretation of the data.

5. The same concern regarding the use of violin plots to correlate gene expression with bivalent genes through development (Figure 6c) as mentioned earlier. It would be better to use DEGs and intersect them. This is particularly important given the wide range of gene expression levels in the already poised state.

In relation to this reviewer's Point 1, we have also performed DEG-based analysis, which is presented in new Figure 6c and Supplementary Figure 2.

We described this point in the revised manuscript (page 10, line 326).

6. The authors limited their analyses to promoter/gene body regions. A survey of the bivalent marks and accessibility at enhancer regions would be also beneficial for understanding the changes at the chromatin landscape through development.

The results presented in Figure 3 demonstrate that chromatin accessibility in the promoter region changes significantly during neuronal differentiation, and this paper has primarily focused on the promoter regions. However, as the reviewer pointed out, we also analyzed the association between DEGs and chromatin accessibility at enhancer regions (new Figure 4A, right panel). Our analysis revealed that most enhancer regions of DEGs remained constitutively open during neuronal differentiation, suggesting that differential accessibility at enhancer regions plays a lesser role in regulating gene expression changes compared to the promoter regions.

We again thank the reviewer for emphasizing the importance of the transition of chromatin accessibility at promoter regions. We described this point in the revised manuscript (page 7, line 211).

7. The authors begin by examining TFs enriched at E16 DHS regions and suggest that TrxG and PcG factors are highly enriched in neurons, initiating their investigation of bivalent marks. However, they later conclude that bivalent marks are present in the NPC state and later become accessible. It is unclear why PRC factors would be enriched at the neuronal stage when the authors conclude that the chromatin becomes more open (potentially by removal of K27me3). The authors should refine this section of the manuscript to better rationalise their methodology and results.

We are grateful to the reviewers for pointing out the insufficient explanation in Figure 6. This section aimed to investigate the mechanism by which open genomic regions at E16.0 were established. To explore this, we used ChIP-atlas to identify the transcription factors enriched in the E16.0 DHSs. Our analysis revealed that many components of Polycomb Group (PcG) proteins, which were previously observed in Pluripotent stem cell types (which share similarities with NPCs), as well as Trithorax Group (TrxG) proteins, which are associated with Neural types, were enriched in these regions. Based on these findings, we hypothesized that the binding of both TrxG and PcG proteins, which characterizes a bivalent state, may play a key role in chromatin opening in NPCs until E16.0. Therefore, we focused on analyzing bivalent genes in NPCs, rather than in E16.0 neurons, as shown in new Figure 6.

We described this point in the revised manuscript (page 10, line 312).

8. Do the authors find any expressional changes of the suggested candidate proteins at the RNA or protein levels through development?

We thank the reviewer for the valuable suggestions. We agree that changes in the expression of TrxG and PcG components during neuronal differentiation are crucial for understanding the mechanism of chromatin structural changes in bivalent genes. In response, we examined the expression levels of genes encoding components of PcG or TrxG, as determined by Schuettengruber et al., *Cell*, 2017, in our RNA-seq dataset (new Supplementary Data 8). Our analysis revealed that more than half of these genes showed significant changes in expression, suggesting that alterations in the activity of PcG, TrxG, or both may contribute to chromatin opening during neuronal differentiation.

We described this point in the revised manuscript (page 14, line 459).

9. The mechanisms driving the activation and expression of poised neuronal genes through the development of deep-layer neurons is not uncovered. The authors suggest certain histone modifiers and the DNA methyltransferase Dnmt3 as potential drivers of chromatin landscape and transcriptional regulation changes; however, this remains speculative, as there is no direct evidence or validation of these factors binding to the identified target regions or changes in DNA methylation states. The authors should provide validation of their candidate factors' presence at potential targets, as well as changes in DNA methylation if they want to conclude these as the mechanisms driving deep-layer neuron development.

We thank the reviewer for highlighting the critical issue of the mechanism underlying the activation of poised genes. To gain mechanistic insight into chromatin opening during neuronal differentiation, we focused on the roles of Dmrt3 and Dmrta2, not Dnmt3, since the DNA motif of Dmrt3 was enriched in the DHSs at E16.0 (new Figure 7a, Supplementary Data 4).

Dmrt3 and its family member Dmrta2 are transcription factors primarily involved in transcriptional repression. Previous studies and our RNA-seq results indicate that they are highly expressed in NPCs and that their expression decreases during neuronal differentiation (new Figure 7B; Desmaris et al., *J. Neurosci.*, 2018; Konno et al., *Development*, 2019). Importantly, previous research suggests that Dmrt3 and Dmrta2 play a role in repressing premature differentiation of NPCs.

Based on these observations, we hypothesized that Dmrt3 and Dmrta2 maintain the closed state of neuronal genes in NPCs, and that their reduction during neuronal differentiation leads to chromatin opening and gene activation. To test this hypothesis, we analyzed the ChIP-seq of Dmrt3 and Dmrta2, as well as RNA-seq of double knockout mice for both *Dmrt3* and *Dmrta2*. Our results showed that Dmrt3 and Dmrta2 specifically bound to the promoters of genes that opened only at E16.0 (new Figure 7C). Moreover, upregulated genes in the double knockout of *Dmrt3* and *Dmrta2* significantly overlapped with the upregulated DEGs during neuronal differentiation (new Figure 7D-E). These findings indicate the association between the reduction of Dmrt3 and Dmrta2 and chromatin opening and gene activation during neuronal differentiation.

We described this point in the revised manuscript (page 11, line 345).

Minor comments:

1. The manuscript would improve with proofreading by a native English speaker.

Thank you for your comment. We would like to inform you that the previous manuscript was proofread by a native English speaker, and we have also ensured that the revised manuscript underwent the proofreading process.

2. The motif analysis can be included in the main figures.

We appreciate the reviewer's positive suggestions. Regarding point 9, we have moved the results of the motif analysis to new Figure 7A. Additionally, based on point 3 of this reviewer, we have included the full motif analysis as Supplementary Data 3-7.

Reviewer #2 (Significance (Required)):

By introducing a novel genetic labelling method that tracks neurons based on their birthdates, the study provides a precise way to examine differentiation *in vivo*, adding valuable insights beyond traditional *in vitro* approaches. The combination of RNA-seq and DNase-seq analyses reveals how chromatin accessibility changes, particularly in bivalent genes, play a crucial role in neuronal maturation. This work highlights the importance of chromatin dynamics in establishing neuronal identity. The techniques and findings provide a useful framework for future studies, offering a path for deeper exploration of chromatin regulation across different neuronal types, stages of development, or disease contexts, making it a valuable contribution to the field of developmental neurobiology.

While the manuscript suggests the involvement of chromatin regulators such as Trithorax and Polycomb proteins, as well as Dnmt3 and DNA methylation, it lacks direct mechanistic evidence, such as ChIP-seq, bisulfite-seq, or loss-of-function experiments, to substantiate these claims. The study focuses exclusively on deep-layer excitatory neurons, without comparisons to other neuronal subtypes or non-neuronal cells. Including such comparisons would help determine whether the observed chromatin changes are unique to this specific population or part of a broader developmental process.

The bioinformatics analyses and interpretations are limited and require further clarification and refinement.

The proposed mechanisms are not fully explored, leaving the manuscript largely descriptive rather than providing a detailed mechanistic understanding.

We would like to thank the reviewer once again for their valuable suggestions to improve our manuscript. By implementing the reanalysis, we have addressed the reviewer's concerns and believe these revisions have enhanced the quality of the paper.

Reviewer #3 (Evidence, reproducibility and clarity (Required)):

In this manuscript the authors use in utero electroporation of tamoxifen inducible reporters to permanently mark cortical neurons with a common birthdate. They then FACS harvest these cells for bulk DNase seq and RNA seq to see changes in chromatin regulation and gene expression as these newborn immature cortical neurons become deep layer neurons. As has been shown in prior studies that have addressed other neuronal types or used different methods to isolate developmental cell stages in the CNS, the authors find correlated changes between the opening or closing of chromatin with changes in gene expression. They use this information to localize chromatin marks that are associated with the differential expression of genes and conclude that many of the differential genes are bivalent for active and repressive chromatin marks. Finally the authors cross this dataset with a microarray they did of BDNF-inducible genes in cortical culture and suggest enrichment of this program in the differentially regulated gene set from in vivo.

Reviewer #3 (Significance (Required)):

The idea that chromatin regulation coordinates developmental changes in gene expression in neurons has been addressed with several different strategies over the past decade including prior strategies that allow for isolation of neurons with common birth dates. Many current strategies (well cited by the authors) use single cell sequencing and computational algorithms to deconvolve differentiation state from complex mixtures. This study takes an alternative approach to experimentally label these developmental stages which is nice to see for the validation of ground truth. However the study does not go far beyond current knowledge to use this method to add new concepts to the field. The main point of innovation seems to be the observation that the newborn neurons are primed at the chromatin level to express deep layer markers at the time they are born during embryonic life. This is useful to see but not unexpected on the basis of large scale single cell datasets. They also show that bivalent promoters prime developmental stage specific gene expression (in addition to the well-established function of this form of regulation in fate determination), however this too has been shown already in other neuron types.

We are very pleased that the reviewer evaluated our method as "nice to see for the validation of ground truth" and distinguished it from the current mainstream approach that traces the differentiation process computationally using single-cell analysis. On the other hand, we also agree with the reviewer's assessment that our previous manuscript did not provide novel insights beyond existing knowledge. As mentioned in our response to Point 9 of Reviewer #2, we have since analyzed the role of Dmrt3 and Dmrt2 in gene expression and chromatin structure during the neuronal differentiation process. This analysis has enabled us to provide new insights into the neuronal differentiation process.

In addition to these conceptual limitations, there are some poorly supported comments in the text. For example, the fact that their microarray shows some genes in a category called "apoptosis" that are BDNF-sensitive does not meaningfully suggest that BDNF induces excitotoxicity in embryonic cortical culture. BDNF has been well established as a survival factor for many kinds of neurons and is a common additive to serum-free media supplements (like B27). The appearance of "apoptosis" terms in the upregulated genes on the microarray more likely suggests either that the microarray is a poor detector of differential gene expression or that the genes in question are inaccurately categorized as "apoptotic" (GO terms are not terribly specific indicators of gene function). If the authors really wanted to test if BDNF was inducing apoptosis their cultures they could test this. However to use only the GO term data in such a strong statement about the biology of their system caused me to question the rigor of either their data or their analysis.

We are grateful to the reviewers for their important comments. We also agree that BDNF is an important neurotrophic factor and do not believe that it induces cell death. Therefore, we checked the following 40 genes, which showed chromatin closing from E12.0 to E16.0 and upregulation upon BDNF stimulation, along with the GO term "programmed cell death":

Cdip1, Diablo, Pla2g6, Braf, Tnfrsf25, Pa2g4, Mcl1, Hpn, Cebpb, EphA2, Plk3, Herpud1, Crip1, Dusp1, Sphk1, Irf5, Bag3, Stil, Fosl1, Cadm1, Lhx3, Hip1r, Relt, Irs2, Bmp8a, Ptcra, Mef2d, Prkcz, Rnf41, Pcid2.

As a result, we found that none of these genes are involved in the main pathways of apoptosis. From this, we understand that the GO terms related to cell death are listed in Figure 5F because "the genes in question are inaccurately categorized as 'apoptotic'," as the reviewer pointed out.

We apologize for the misleading discussion in the previous manuscript and would like to thank the reviewer again for highlighting this important point. We have corrected this in the new manuscript (page 9, line 304).

A second example is the section about promoters being the focus of their discussion for DHS sites. Sure figure 3c shows promoters are more likely to be open compared with their contribution to the genome overall, but this is entirely expected since they are major gene TF binding sites, which is what DNase detects. However promoters do not look to be more likely to be differentially regulated over time (3c vs 3e), and the statement that promoters are more enriched in opening compared with closing sites would require a statistical statement. Distal DHS sites appear equally more abundant in opening sites too.

We thank the reviewer for their thoughtful comments on our results. As the reviewer points out, the proportion of promoter regions in the opening DHS in Figure 3e is not as high compared to that in Figure 3c. However, as described in the Abstract and Introduction sections, our main interest lies in understanding how neurons acquire their function during the differentiation process, and we focused primarily on comparing neuron-specific and NPC-specific DHS. In the comparison within Figure 3E, it is evident that the opening DHS has a higher proportion of promoter regions than the closing DHS. We have made the necessary revisions to clarify this point and avoid any misunderstanding (page 7, line 211).

On the other hand, we agree with the reviewer that distal DHSs may also play important roles in regulating neuronal differentiation. Therefore, in the revised manuscript, we also analyzed the association between DEGs and chromatin accessibility at enhancer regions (new Figure 4A, right panel), as mentioned in our response to Point 6 of Reviewer #2. Our analysis revealed that most of the enhancer regions of DEGs were constitutively open during neuronal differentiation, suggesting that the role of the transition of chromatin accessibility at enhancer regions in regulating gene expression change might be less significant than at promoter regions. Again, we thank the reviewer for emphasizing the importance of the transition of chromatin accessibility at promoter regions.

We described this point in the revised manuscript (page 7, line 208).

Third decision letter

MS ID#: dev.204564R2

MS Title: In vivo transition in chromatin accessibility during differentiation of deep-layer excitatory neurons in the neocortex

Authors: Seishin Sakai; Yurie Maeda; Mai Saeki; Daijiro Konno; Keita Kawaji; Fumio Matsuzaki; Yutaka Suzuki; Yukiko Gotoh; Yusuke Kishi

Article Type: Review Commons Transfer

Dear Dr Kishi,

I have now received all the referees reports on the above manuscript, and have reached a decision. The referees' comments are appended below, or you can access them online: please go to .

The overall evaluation is positive and we would like to publish a revised manuscript in Development, provided that the referees' comments can be satisfactorily addressed. Referee 2 would like more discussion regarding the correlation between enhancer/promoter changes and gene expression. I agree that this would be helpful to the readers. Please attend to all of the reviewers'

comments in your revised manuscript and detail them in your point-by-point response. If you do not agree with any of their criticisms or suggestions explain clearly why this is so. If it would be helpful, you are welcome to contact us to discuss your revision in greater detail. Please send us a point-by-point response indicating your plans for addressing the referees' comments, and we will look over this and provide further guidance.

Comments from the Reviewers:

Reviewer 1: COMMENTS ON TEXT

COMMENTS ON DISPLAY ITEMS

The authors have made significant improvements in the revised manuscript by incorporating more comprehensive and updated data analyses. They have effectively addressed my concerns, enhancing the clarity and robustness of the study. The addition of supplementary tables further strengthens the manuscript, providing valuable context and detailed information that support the main findings.

Overall, the revised manuscript meets the standards for publication and is suitable for acceptance in its current form.

Reviewer 2: COMMENTS ON TEXT

The authors have offered a strong response to the concerns of the reviewers. The story is now much more clear, errors have been corrected, and the addition of the new analyses of the Dmrts is interesting. The data and the manuscript will be of interest to developmental neurobiologists who study chromatin regulation in brain development.

I have only one remaining concern, and that regards the discussion of promoters versus enhancers. In my opinion a more detailed description of the data from Figure 3E and Figure 4A in the text and a more nuanced conclusion will help the reader interpret these data. Only text revisions are required.

I agree with the authors that more promoters open than close in Figure 3E, though that does not answer the question of whether promoters or enhancers are more important for neuronal differentiation. What matters for the function of the promoters is that the authors have shown that opening promoters correlate with increases in gene expression. This all makes sense and it is nicely done. Opening promoters definitely drive neuronal gene expression.

However Figure 3E also shows that about 80% of the sites that open between E12 and E16 are NOT promoters. Based on DHS data from other papers (such as Heintzman 2009) these are likely to be enhancers. Given that these non-promoter sites account for 80% of the developmentally opening sites, it seems that the authors should comment on this in the text.

The authors choose to emphasize that they do not find a strong correlation between opening at non-promoter sites and increases in gene expression, compared with what they saw for promoters (though the data on the left and right side of Figure 4A are not statistically compared - they are only qualitative visual comparisons, so it is hard to relate one to the other). However I have two concerns - one is about how enhancers are defined and the other is how their targets are called.

First, the easiest way to determine if the non-promoter opening sites are developmental enhancers would be to see if they overlap with H3K27ac at E16 but not at E12. This would suggest that as they open, then they gain enhancer marks. The authors may not have access to these data, and it is reasonable to consider gathering these data to be beyond the scope of the study. Instead the authors use a database that categorically calls some subset (?) of these sites as enhancers. It was unclear to me from a glance at the reference how those calls were made or if the database shows changes in enhancer marks at these opening sites over this same timecourse. Because this is important for the interpretation of these data, the authors should add a description of the comparison between their opening sites in Figure 3E and their called enhancers in Figure 4A to help the reader know which sites are being evaluated.

Second, it is hard to map enhancers to genes - sometimes they regulate the closest gene and sometimes they do not. The authors should describe how they mapped their called enhancers to genes for figure 4A.

COMMENTS ON DISPLAY ITEMS

1) In the figures many of the colors are not explained either with legend in the figure or in the text of the figure legend. For example in Figure 3B the first bar is red and the rest are green. In Figure 3D there are red and blue violin plots, but it is not clear in the figure what these represent.

2) For the DNase data in particular, the story would be stronger if the authors showed at least a few clear examples of genes that change accessibility at their promoters or enhancers, rather than only showing the idealized drawing in figure 4A. As a reader I always like to see a least some real data where a statistical change is being called to understand how robust the calls are.

3) Why is the value of the heat map 1-20 in the promoters of figure 4A and 1-50 in the enhancers? This makes the enhancers seem to correlate less with transcription than the promoters. The heat maps should be the same for this kind of visual comparison.

Third revision

Author response to reviewers' comments

Reviewer 2: COMMENTS ON TEXT

The authors have offered a strong response to the concerns of the reviewers. The story is now much more clear, errors have been corrected, and the addition of the new analyses of the Dmrts is interesting. The data and the manuscript will be of interest to developmental neurobiologists who study chromatin regulation in brain development.

We really appreciate this reviewer's positive comments on our revision.

I have only one remaining concern, and that regards the discussion of promoters versus enhancers. In my opinion a more detailed description of the data from Figure 3E and Figure 4A in the text and a more nuanced conclusion will help the reader interpret these data. Only text revisions are required.

I agree with the authors that more promoters open than close in Figure 3E, though that does not answer the question of whether promoters or enhancers are more important for neuronal differentiation. What matters for the function of the promoters is that the authors have shown that opening promoters correlate with increases in gene expression. This all makes sense and it is nicely done. Opening promoters definitely drive neuronal gene expression.

However Figure 3E also shows that about 80% of the sites that open between E12 and E16 are NOT promoters. Based on DHS data from other papers (such as Heintzman 2009) these are likely to be enhancers. Given that these non-promoter sites account for 80% of the developmentally opening sites, it seems that the authors should comment on this in the text.

We sincerely thank the reviewer for highlighting the importance of enhancer regions. As the reviewer noted, we do not intend to exclude the contribution of enhancer regions in the early neuronal differentiation process, but we recognize that our previous manuscript may have been misleading in this regard. Therefore, we have revised the Discussion section to emphasize the potential contribution of enhancer regions, explicitly stating that 79% of the regions that open from E12.0 to E16.0 are located in non-promoter regions, as the reviewer pointed out (new page 13, line 404; page 14, line 412).

The authors choose to emphasize that they do not find a strong correlation between opening at non-promoter sites and increases in gene expression, compared with what they saw for promoters (though the data on the left and right side of Figure 4A are not statistically compared - they are only qualitative visual comparisons, so it is hard to relate one to the other). However I have two concerns - one is about how enhancers are defined and the other is how their targets are called.

First, the easiest way to determine if the non-promoter opening sites are developmental enhancers would be to see if they overlap with H3K27ac at E16 but not at E12. This would suggest that as they open, then they gain enhancer marks. The authors may not have access to these data, and it is reasonable to consider gathering these data to be beyond the scope of the study. Instead the authors use a database that categorically calls some subset (?) of these sites as enhancers. It was unclear to me from a glance at the reference how those calls were made or if the database shows changes in enhancer marks at these opening sites over this same timecourse. Because this is important for the interpretation of these data, the authors should add a description of the comparison between their opening sites in Figure 3E and their called enhancers in Figure 4A to help the reader know which sites are being evaluated.

We appreciate the reviewer for highlighting the lack of clarity regarding the definition of enhancers in our previous manuscript. As the reviewer mentioned, the most definitive approach to identifying developmental enhancers would be to perform ChIP-seq for enhancer marks, such as H3K27ac and H3K4me1, across all relevant cell types in this study, including E12.0 NPCs, E13.0, E14.0, and E16.0 neurons. However, conducting such experiments is beyond the scope of the present study. Therefore, we relied on the public enhancer database, Enhancer-Atlas V2, which provides enhancer region annotations for many cell types (Gao and Qian, *Nucleic Acids Res.*, 2019). In our analysis, we used the integrated enhancer list for NPCs and NeuronCortical, as our datasets include both undifferentiated E12.0 NPCs and postmitotic E16.0 neurons.

We agree that the method used to define enhancer regions is critical, as the reviewer pointed out. To address this, we have added a brief description of how enhancer regions are defined in Enhancer-Atlas V2 in the Materials and Methods section of the revised manuscript (new page 19, line 615). Briefly, enhancer regions were identified by integrating public data from the following three sources: (1) epigenetic modifications, using ChIP-seq for histone modifications and transcription factors (TFs) specific to each cell type; (2) chromatin accessibility, assessed by DNase-seq, formaldehyde-assisted isolation of regulatory elements (FAIRE-seq), and ATAC-seq; and (3) eRNA expression, analyzed using Pol II ChIP-seq, genomic run-on (GRO-seq), and cap analysis of gene expression (CAGE).

Furthermore, we compared the opening sites identified in Fig. 3E with the enhancer regions annotated in Enhancer-Atlas V2 as shown in Fig. 4A, as this reviewer proposed, and found that a significant proportion (19.7%) of non-promoter opening sites in Fig. 3E overlapped with enhancer regions. We have described this comparison in the Discussion section (new page 14, line 415) to help readers interpret which sites are being evaluated in our study.

Second, it is hard to map enhancers to genes - sometimes they regulate the closest gene and sometimes they do not. The authors should describe how they mapped their called enhancers to genes for figure 4A.

Related to the above point, the method used to map enhancers to genes is also important. For this, we again utilized the Enhancer-Atlas V2 database. In this database, enhancer-gene pairs are determined using the EAGLE software, which integrates correlations between enhancer activity, gene expression levels, and the distance between enhancers and genes (Gao and Qian, *PLoS Comput. Biol.*, 2019). We have also briefly described this method in the Materials and Methods section of the revised manuscript (new page 20, line 621).

COMMENTS ON DISPLAY ITEMS

1) In the figures many of the colors are not explained either with legend in the figure or in the text of the figure legend. For example in Figure 3B the first bar is red and the rest are green. In Figure 3D there are red and blue violin plots, but it is not clear in the figure what these represent.

We used consistent colors to represent each cell type throughout the figures. E12.0 NPCs are shown in red and neurons in green in the revised Figs. 2B, 3B, 4D, 5C, 6D, and 7E. To clarify this color

usage, we updated the color schemes in revised Figs. 1A and 2A. As suggested by the reviewer, we also applied the same color scheme to all data in revised Fig. 3D.

2) For the DNase data in particular, the story would be stronger if the authors showed at least a few clear examples of genes that change accessibility at their promoters or enhancers, rather than only showing the idealized drawing in figure 4A. As a reader I always like to see at least some real data where a statistical change is being called to understand how robust the calls are.

We thank the reviewer for this helpful suggestion to improve our presentation. In response, we have added an example of the *Rbfox1* gene, which is highly expressed in neurons (Gehman et al., *Nat. Genet.*, 2011), acquires DHSs only at E16.0, and shows increased expression levels during neuronal differentiation (new Fig. 4C). This provides a representative case of a gene with changes in chromatin accessibility and expression, as requested.

3) Why is the value of the heat map 1-20 in the promoters of figure 4A and 1-50 in the enhancers? This makes the enhancers seem to correlate less with transcription than the promoters. The heat maps should be the same for this kind of visual comparison.

In the previous version of the manuscript, we used different value scales for promoters and enhancers in Fig. 4A because direct comparison between these two groups was difficult due to differences in their characteristics, as the reviewer pointed out in the above comment. However, we understand the reviewer's concern, and, in the revised manuscript, we have unified the value scale to 1-20 for both promoters and enhancers in the new Fig. 4A to facilitate direct comparison, as suggested.

Fourth decision letter

MS ID#: dev.204564R3

MS TITLE: In vivo transition in chromatin accessibility during differentiation of deep-layer excitatory neurons in the neocortex

AUTHORS: Seishin Sakai; Yurie Maeda; Mai Saeki; Daijiro Konno; Keita Kawajji; Fumio Matsuzaki; Yutaka Suzuki; Yukiko Gotoh; Yusuke Kishi

ARTICLE TYPE: Review Commons Transfer

I am happy to tell you that your manuscript has been accepted for publication in Development, pending our standard ethics checks.